# EvolveR: Self-Evolving LLM Agents through an Experience-Driven Lifecycle

## Abstract

Current Large Language Model (LLM) agents show strong performance in tool use, but lack the crucial capability to systematically learn from their own experiences. While existing frameworks mainly focus on mitigating external knowledge gaps, they fail to address a more fundamental limitation: the inability to iteratively refine problem-solving strategies. In this work, we introduce **EvolveR**, a framework designed to enable agent to self-improve through a complete, closed-loop experience lifecycle. This lifecycle comprises two key stages: (1) **Offline Self-Distillation**, where the agent's interaction trajectories are synthesized into a structured repository of abstract, reusable strategic principles; (2) **Online Interaction**, where the agent interacts with tasks and actively retrieves distilled principles to guide its decision-making, accumulating a diverse set of behavioral trajectories. This loop employs a policy reinforcement mechanism to iteratively update the agent based on its performance. We demonstrate the effectiveness of EvolveR on complex multi-hop question-answering benchmarks, where it achieves superior performance over strong agentic baselines. Our work presents a comprehensive blueprint for agents that learn not only from external data but also from the consequences of their own actions, paving the way for more autonomous and continuously improving systems.

## 1 Introduction

Large Language Models (LLMs) have driven the development of autonomous agents capable of solving diverse tasks through advanced reasoning and tool use (Shen et al., 2023; Luo et al., 2025; ang Gao et al., 2025). However, a significant limitation emerges when these agents engage in sequential tasks: each interaction is treated independently. They approach tasks as isolated episodes, suffering from operational amnesia and failing to learn from past successes or avoid prior mistakes(Yao et al., 2023b). This inability to leverage experience fundamentally hinders their development toward greater autonomy and intelligence.

Humans, by contrast, learn through a continuous lifecycle, leveraging both successes and failures to refine strategies over time (Flesch et al., 2018). For example, a student solving math problems reflects on recurring errors and successful approaches to extract general problem-solving strategies. This cycle of interaction, reflection, and abstraction is the cornerstone of developing expertise (Anderson, 1993). Endowing LLM agents with a comparable lifecycle is the key to bridging the gap between episodic problem-solving and sustainable self-improvement. While existing frameworks like Retrieval-Augmented Generation (RAG) effectively address knowledge gaps, they fail to solve a more fundamental limitation: the agent's inability to systematically learn from the consequences of its own interactions (Yan et al., 2025).

As Figure 1 shows, prior works have attempted to address this limitation, but with critical shortcomings. Researchers store natural language reflections across tasks with a powerful external LLM in an external memory (Zhao et al., 2024; Zhou et al., 2025). While resource-efficient, this approach treats such reflections as a transient hint, leaving the agent's intrinsic policy unchanged. On the other hand, learning by recalling raw cases retrieves entire past trajectories to directly guide decision-making. However, this reliance on raw cases struggles to generalize and, more importantly, fails to abstract. The agent merely mimics past solutions instead of distilling the reusable strategic principles that made them successful (Chen et al., 2023).

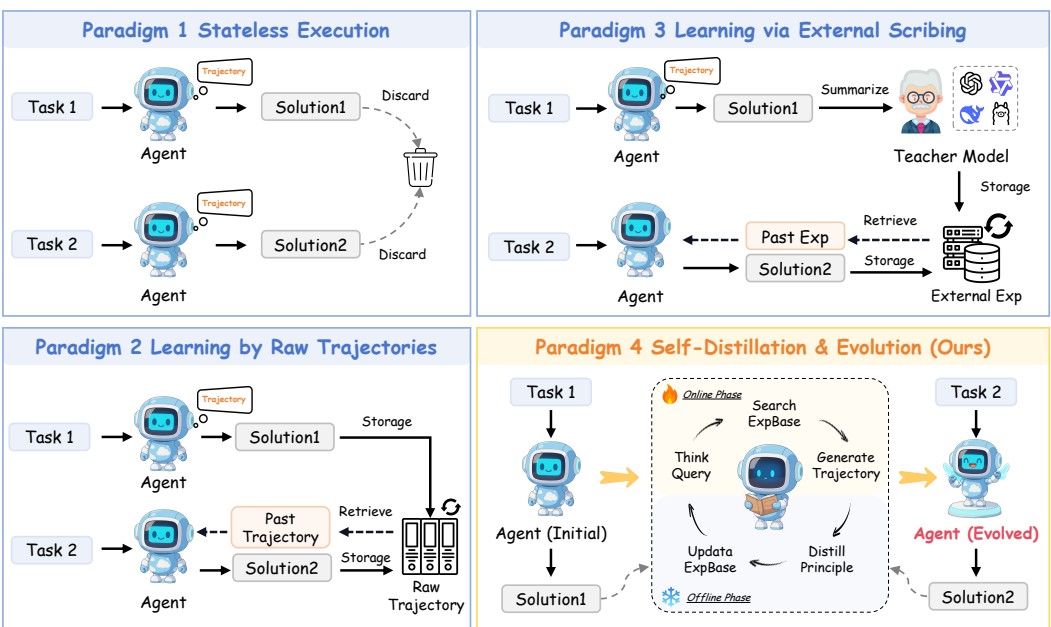

Figure 1: An illustration of four major paradigms for LLM agent learning. (1) **Stateless Execution**: Standard agents discard experiences after each task; (2) **Learning by Raw Trajectories**: Agents retrieve raw, un-distilled past trajectories; (3) **Learning via External Scribing**: Agents rely on an external teacher model to distill insights; (4) **EvolveR (Ours)**: A complete, self-contained lifecycle where the agent autonomously distills its own experiences into principles and evolves its policy.

To overcome these challenges, we introduce EvolveR, a framework that enables agents to self-evolve by utilizing their own experiences. EvolveR implements a full experience lifecycle, in which agents collect trajectories through Online Interaction, distill them into a library of abstract strategic principles during Offline Self-Distillation, and subsequently learn to apply these principles to new tasks. Crucially, EvolveR completes the experience lifecycle with a reinforcement learning mechanism that enables the agent to utilize experience. The agent does not merely mimic its past interactions; it evolves based on what it has learned. EvolveR maintains a dynamic experience base where newly distilled principles are semantically deduplicated and continuously evaluated via a metric score that tracks historical effectiveness.

We demonstrate EvolveR's effectiveness on complex question-answering benchmarks, where it significantly outperforms strong agentic baselines. Our contributions can be summarized as follows:

- **We propose the Experience-Driven Self-Evolution Paradigm, a novel, closed-loop lifecycle for LLM agents.** In contrast to agents that forget past interactions, EvolveR systematically integrates a complete cycle of *online interaction*, *offline experiences self-distillation* and *policy evolution*. This process enables the agent to continuously transform raw trajectories into a curated repository of strategic principles, establishing a foundation for adaptive agents.

- **We introduce a complete system for dynamic experiences curation.** This system goes far beyond simple experience storage. It features: (1) a **self-distillation** mechanism, where the agent autonomously distills principles from previous interactions; and (2) a full **maintenance pipeline**, including semantic deduplication, integration, and quality control guided by a dynamic metric score.

- **We provide extensive empirical validation of the EvolveR paradigm across multiple model scales.** Our experiments on a diverse suite of complex QA benchmarks demonstrate the effectiveness of our approach. Detailed ablation studies confirm that the synergy of our proposed curation and self-distillation mechanisms is critical to the framework's success, revealing a key insight: while the self-distillation mechanism is less effective on smaller-scale models, it **surpasses distillation by a stronger, external teacher model** at the 3B scale, validating the importance of cognitive alignment.

## 2 RELATED WORK

### 2.1 CONTINUAL LEARNING AND SELF-EVOLVING AGENTS

Continual learning (CL) aims to enable models to learn sequentially while mitigating catastrophic forgetting (Parisi et al., 2019; Wang et al., 2024). While various replay-based and regularization methods have been proposed, most CL paradigms assume predefined task boundaries and focus on knowledge preservation rather than active acquisition in open-ended environments (Kirkpatrick et al., 2017; Ding et al., 2024; Huai et al., 2025a;b). The pursuit of self-evolving agents moves beyond these limitations by enabling systems to grow autonomously from experience. Frameworks such as Reflexion and Generative Agents explore self-improvement through self-play and reflective reasoning, often storing past trajectories as memory to guide future actions (Shinn et al., 2023; Wei et al., 2022; Yao et al., 2023a; Besta et al., 2024; Yao et al., 2023b). However, these systems either store raw, unstructured data or rely on memory mechanisms that are not designed for the systematic, long-term distillation and refinement of abstract strategic knowledge. Instead of relying on external data streams, our agent autonomously generates and refines its own experiences through an iterative cycle of online interaction and offline reflection.

### 2.2 LLM AGENTS AND REINFORCEMENT LEARNING

LLM agents have been widely explored through frameworks such as ReAct, which interleaves reasoning and actions, and Reflecion, which improves task performance via self-reflection (Yao et al., 2023b; Shinn et al., 2023). While these approaches are primarily prompt-based and stateless, they prevent long-term accumulation of strategic knowledge. External memory frameworks like ExpeL address this limitation by reusing past trajectories, but they do not enable systematic self-improvement across tasks (Zhao et al., 2024). While effective, these methods often rely on simple prompting and are inherently stateless, limiting their ability to internalize knowledge across tasks. Recent work has increasingly turned to reinforcement learning (RL) to train agents for long-horizon, multi-turn tasks. However, applying RL is challenging due to sparse rewards and the need for stable training signals. Search-R1 (Jin et al., 2025), O2-Searcher (Mei et al., 2025), and AutoRefine (Shi et al., 2025) all use RL to train LLMs to generate and interact with external search tools. While these works successfully optimize the LLM's interaction with external factual knowledge, they do not address the broader challenge of an agent's self-improvement through its own internal experience.

## 3 METHOD

In this section, we present **EvolveR**, a novel framework designed to enable agent self-evolution through a complete, closed-loop experience lifecycle. Inspired by the human cycle of work and reflection, our approach is structured around three core, interconnected components, as depicted in Figure 2. First, in the **Offline Experience Self-Distillation** phase, the agent's policy parameters are frozen, and it systematically distills raw trajectories into a curated base of strategic principles. Second, during the **Online Interaction** phase, the agent applies this distilled wisdom to guide its deliberative reasoning and action, generating new, high-quality interaction data. Finally, the entire cycle is driven by a **Policy Evolution** mechanism, where the trajectories collected online are used to update the agent's policy parameters via reinforcement learning, thus closing the loop. This iterative process allows the agent to continuously transform its interactions into evolving expertise.

### 3.1 PRELIMINARIES: FORMALIZING AGENT INTERACTION

At each state $t$, the agent, situated in an unknown state $s_t$, selects an action $a_t \in \mathcal{A}$ based on its policy. Our agent's action space $\mathcal{A}$ is designed for complex, knowledge-intensive tasks and comprises three key operations:

- <search_experience>: Agent queries its internal experience base $\mathcal{E}$ to retrieve relevant principles distilled from past trajectories. Environment returns retrieved principles as an observation.
- <search_knowledge>: Agent queries an external knowledge base (e.g., a search engine) to acquire factual information. Environment returns retrieved information as an observation.
- <answer>: Agent outputs its final answer to the problem and concludes the interaction.

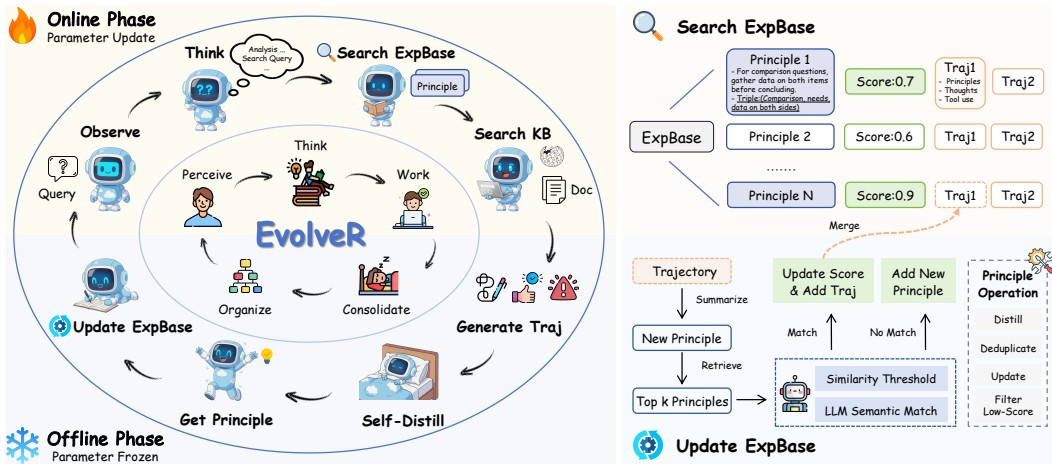

Figure 2: **Overview of the EvolveR framework's experience lifecycle. Left**: The main loop alternates between an *Online Phase*, where the agent interacts with the environment and its policy parameters are updated via RL, and an *Offline Phase*, where the agent's parameters are frozen and it performs self-distillation and maintains its Experience Base ($\mathcal{E}$). **Top Right**: A detailed view of the `Search ExpBase` action, where the agent retrieves scored principles along with their associated trajectories. **Bottom Right**: The `Update ExpBase` process, which involves summarizing trajectories and applying a suite of curation operations (distill, deduplicate, update, and filter).

## 3.2 THE EVOLVER LIFECYCLE: FROM INTERACTIONS TO PRINCIPLES

### 3.2.1 OFFLINE EXPERIENCE SELF-DISTILLATION

The core of EvolveR is a self-perpetuating lifecycle designed to transform raw interaction data into a strategic principle. This process is divided into two distinct, alternating phases: an offline self-distillation phase for distilling the principle, and an online interaction phase for applying the principle and gathering new interaction data.

**Principle from Self-Distillation.** The process begins with self-distillation. We leverage the agent's own policy model $\pi_\theta$ to analyze its past interaction trajectories. By adopting the persona of an expert through carefully designed prompts, the model reviews each trajectory and, based on its outcome, distills the core strategic insight into a concise natural language statement. This results in either a **guiding principle** from a success or a **cautionary principle** from a failure.

Inspired by structured memory frameworks such as Mem0 (Chhikara et al., 2025) and G-Memory (Zhang et al., 2025), each principle consists of two components: a natural language description paired with several structured knowledge triples, as illustrated in Figure 2. This self-distillation approach enables the agent to autonomously generate reusable knowledge.

**Deduplication and Integration.** To maintain a high-quality experience base ($\mathcal{E}$), we do not add every distilled principle. Instead, each new principle undergoes a rigorous integration process. First, to handle redundancies arising from similar trajectories (e.g., from GRPO sampling), we perform a deduplication step. We use the agent model $\pi_\theta$ to pair-wise check for semantic equivalence among newly generated principles that originate from the same problem, keeping only one representative from each semantically equivalent cluster.

Second, for each unique principle, we apply a two-stage matching procedure: we first retrieve the most similar existing principles from $\mathcal{E}$ via embedding similarity, then prompt the agent model to provide a binary semantic equivalence judgment. If a principle is novel, it is added as a new entry in $\mathcal{E}$; otherwise, the new trajectory is merged under the existing principle, enriching it without introducing redundancy.

Let $p_{\text{cand}}$ be a new candidate principle distilled from trajectory $\tau_{\text{src}}$. We update the experience base $\mathcal{E}$ as follows:

$$\mathcal{E} \leftarrow \begin{cases} \mathcal{E} \cup \{p_{\text{cand}}\} & \text{if } \max_{p \in \mathcal{E}} \text{sim}(p_{\text{cand}}, p) < \theta_{\text{sim}} \\ \text{Merge}(\mathcal{E}, p^*, \tau_{\text{src}}) & \text{otherwise} \end{cases} \quad (1)$$

where $\text{sim}(\cdot, \cdot)$ is the cosine similarity between principle, $\theta_{\text{sim}}$ is a similarity threshold, and $p^* = \text{argmax}_{p \in \mathcal{E}} \text{sim}(p_{\text{cand}}, p)$. The Merge operation links $\tau_{\text{src}}$ to its best match $p^*$.

This two-level check ensures that $\mathcal{E}$ grows with novel insights while strengthening existing ones with new evidence.

**Quality Control via Dynamic Scoring.** As the experience base accumulates principles over time, it becomes essential to evaluate their practical utility and prioritize the most effective strategies. To this end, each principle tracks its usage and success counts, enabling the computation of an empirical score that reflects historical performance. We quantify the empirical utility of each principle using a metric score, which is updated as:

$$s(p) = \frac{c_{\text{succ}}(p) + 1}{c_{\text{use}}(p) + 2} \quad (2)$$

where $c_{\text{succ}}(p)$ and $c_{\text{use}}(p)$ are the success and usage counts for a given principle $p$, $s(p)$ is the metric score.

This score provides a reliable measure of a principle's historical effectiveness. To ensure the long-term health of the experience base, we periodically prune principles whose scores fall below a threshold $\theta_{\text{prune}}$. This systematic process of distillation, integration, and quality control ensures that the agent's wisdom remains a compact and high-quality repository of its most effective strategies.

### 3.2.2 ONLINE INTERACTION

Before the online interaction begins, we include a brief cold-start phase to ensure that the model can produce well-formed interaction formats (e.g., <think> and <search_experience>). This phase provides no task knowledge or strategic guidance.

With the format stabilized, the online phase serves as the interactive testbed where the agent applies its distilled principles to solve problems. The agent operates within a deliberative reasoning loop (e.g., Think-Act-Observe), which enables it to engage in multi-turn, autonomous tool use. However, the core novelty of EvolveR's online phase is not the loop itself, but how the principles retrieved from the experience base ($\mathcal{E}$) fundamentally alter the agent's behavior within it.

**Experience as a Strategic Principle.** Unlike standard agents that must discover reasoning patterns from scratch through trial and error, an EvolveR agent is guided by a strategic wisdom provided by its own past experiences. At any point in its reasoning loop, the agent can issue a <search_experience> action. The retrieved principles $\mathcal{P}_k$ do not merely provide factual information; they offer heuristic guidance that shapes the agent's subsequent reasoning. For instance, retrieving a principle such as "For comparison questions, gather data on both items before concluding," can directly influence the agent's internal monologue (<think>) and steer its subsequent potential <search_knowledge> actions. This makes the agent's exploration more efficient and less prone to common pitfalls, as it learns to follow the wisdom in its own distilled principles.

**Generating High-Quality Trajectories for Future Distillation.** The ultimate purpose of the online phase, within the EvolveR paradigm, extends beyond solving the immediate task. It is responsible for generating high-quality data for the next cycle of offline reflection. Because the agent's actions are guided by proven principles, the resulting trajectories, $\tau_{\text{new}}$, are not random walks but are instead rich recordings of structured, experience-guided problem-solving. These trajectories capture the interplay between distilled principles, internal reasoning, and external tool use (e.g., <search_knowledge>), and serve as valuable input for the offline phase, enabling EvolveR to refine existing principles and discover more effective strategies in a virtuous cycle.

### 3.3 POLICY EVOLUTION: CLOSING THE LOOP WITH REINFORCEMENT LEARNING

To enable the agent to learn from its actions and evolve its policy $\pi_\theta$, we employ a reinforcement learning framework. The learning process is guided by a composite reward function and a policy optimization algorithm that leverages the trajectories collected during the online phase.

**Reward Function.** We design a composite reward function $R(\tau)$ for a given trajectory $\tau$ that balances task success with procedural correctness. It is a weighted sum of an outcome reward and a format reward: $R(\tau) = w_o R_{\text{outcome}}(\tau) + w_f R_{\text{format}}(\tau)$.

- **Outcome Reward** $R_{\text{outcome}}$, is a sparse, binary reward based on the final answer's correctness. Following prior work, it is determined by an exact match with the ground truth:

$$R_{\text{outcome}}(\tau) = \text{EM}(a_{\text{pred}}, a_{\text{gold}}) \tag{3}$$

  where $a_{\text{pred}}$ is the final answer extracted from the trajectory $\tau$ and $a_{\text{gold}}$ is the ground truth answer.

- **Format Reward** $R_{\text{format}}$, is a dense shaping reward that evaluates the quality of the reasoning process. Let $N_{\text{think}}(\tau)$, $N_{\text{exp}}(\tau)$ and $N_{\text{know}}(\tau)$ denote the counts of valid <think>, <search_experience> and <search_knowledge> actions within $\tau$. $R_{\text{format}}$ is composed of a think score $R_{\text{think}}$, rewarding a balanced number of reasoning steps, and a search score $R_{\text{search}}$ promoting search experience and knowledge. The final format reward is calculated as:

$$R_{\text{format}}(\tau) = \mathbb{I}(\tau_{\text{complete}}) \cdot \frac{R_{\text{think}}(\tau) + R_{\text{search}}(\tau)}{2} \tag{4}$$

  where $\mathbb{I}(\tau_{\text{complete}})$ is an indicator function that is 1 only if the trajectory contains at least one of each required action type (<think>, any search, and <answer>), and 0 otherwise. This ensures that only structurally complete trajectories receive a format reward.

**Policy Optimization.** The policy $\pi_\theta$ is updated using the collected trajectories. We utilize Group Relative Policy Optimization (GRPO) (Shao et al., 2024), which balances the optimization stability and efficiency by using the average reward of multiple sampled trajectories as a baseline, thus avoiding the need for a learned value function. Specifically, for each input, we sample a group of $G$ trajectories. The policy is then optimized by maximizing the following objective function:

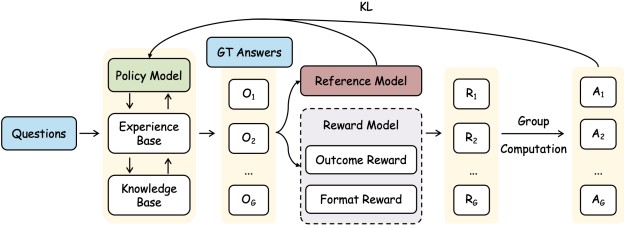

Figure 3: Policy model update optimization algorithm of EvolveR.

$$\mathcal{J}_{\text{GRPO}}(\theta) = \mathbb{E}_{\tau \in \mathcal{D}} \left[ \sum_{t=1}^{|\tau|} \min\left( \rho_t(\theta)\hat{A}_t, \text{clip}(\rho_t(\theta), 1-\epsilon, 1+\epsilon)\hat{A}_t \right) - \beta D_{\text{KL}}[\pi_\theta || \pi_{\text{ref}}] \right] \tag{5}$$

where $\rho_t(\theta) = \frac{\pi_\theta(a_t|h_t)}{\pi_{\text{old}}(a_t|h_t)}$ is the importance sampling ratio, $\hat{A}_t$ is the advantage estimate, and the final term is a KL-divergence penalty to constrain policy updates.

Crucially, this optimization process is deeply integrated with our experience lifecycle. As the agent's actions during the online phase are conditioned on the principles $\mathcal{P}_k$ retrieved from its experience base, the trajectories collected in $\mathcal{D}$ are inherently experience-guided. Consequently, the GRPO update does not merely learn a generic reasoning policy. Instead, it explicitly learns a policy of how to effectively utilize its own distilled wisdom to generate successful outcomes. The optimization process, therefore, reinforces the valuable connections between retrieving high-quality principles and producing high-reward trajectories, successfully closing the learning loop.

## 4 EXPERIMENTS

### 4.1 EXPERIMENTAL IMPLEMENTATION DETAILS

#### 4.1.1 TASKS AND DATASETS

To comprehensively evaluate the EvolveR paradigm, we assess its performance on seven question-answering benchmarks, encompassing both in-domain and out-of-domain datasets. Following prior

work (Jin et al., 2025; Mei et al., 2025), the in-domain datasets, whose training splits are used to build the experience base, include Natural Questions (NQ) (Kwiatkowski et al., 2019) and the multi-hop benchmark HotpotQA (Yang et al., 2018). The out-of-domain datasets, used exclusively for evaluating generalization, encompass the general QA benchmarks TriviaQA (Joshi et al., 2017) and PopQA (Mallen et al., 2022), as well as the more complex multi-hop challenges 2WikiMulti-HopQA (Ho et al., 2020), Musique (Trivedi et al., 2022b), and Bamboogle (Press et al., 2022).

### 4.1.2 BASELINE METHODS

Following prior works, we compare against a comprehensive suite of baselines built upon the Qwen2.5 foundational models. The baselines represent three primary paradigms. First, prompting-based methods, which require no parameter updates, include Direct Inference, Chain-of-Thought (CoT) (Wei et al., 2022), Retrieval-Augmented Generation (RAG) (Lewis et al., 2020), and advanced variants like IRCoT (Trivedi et al., 2022a) and Search-o1 (Li et al., 2025). Second, Supervised Fine-Tuning (SFT) methods represent approaches that learn from static expert data, including standard SFT (Chung et al., 2024) and Rejection Sampling (Ahn et al., 2024). Finally, the most direct competitors are RL methods, against which we benchmark extensively. This category is primarily composed of Search-R1 (Jin et al., 2025), DeepSeek-R1 (Guo et al., 2025), which are also trained with trajectory-level feedback. Specifically, DeepSeek-R1 performs reasoning and answer steps without a search engine, whereas Search-R1 incorporates an external local or web search engine. Together, these baselines provide a challenging evaluation landscape for our proposed paradigm.

### 4.1.3 EVALUATION METRICS

To ensure a direct and fair comparison with prior work in our main results, our primary evaluation metric is Exact Match (EM), a strict measure that requires the predicted answer to exactly match the ground truth after standard normalization. We also report the F1 Score in the analysis of model scales' generalizability, which provides a more comprehensive and robust measure of performance, particularly since ground truths may contain multiple valid answers or aliases.

### 4.1.4 IMPLEMENTATION DETAILS

Our experiments are conducted on the Qwen2.5 model family. Inspired by DeepSeek-R1 (Guo et al., 2025), we introduce a cold-start stage to stabilize early RL training by first fine-tuning the base model on a small, curated dataset of CoT interaction trajectories. Following the setup of Search-R1, we construct this dataset from approximately 700 samples from the NQ and HotpotQA training sets. We utilize the LLama_Factory (Zheng et al., 2024) to fine-tune the model with LoRA. For the agent evolution phase, we employ GRPO for optimization. At each RL step, we sample a batch of 128 prompts, generating $G = 8$ trajectories for each. The agent is then updated, again using Adam, but with a reduced learning rate of $1 \times 10^{-6}$, a warm-up step of 20 and a mini-batch size of 128. All training is conducted on 8 A100 GPUs, leveraging the Verl framework [1] for efficient implementation. We will show more details in Appendix 4.1.

### 4.2 MAIN RESULTS

The main results of our evaluation are presented in Table 1. Our analysis focuses on the comprehensive evaluation conducted on the Qwen2.5-3B and 7B models (we will show more results of different model scales in the 5.1). EvolveR achieves superior average scores 0.382 for 3B and 0.417 for 7B, outperforming all baselines, including strong RL agents like Searcher-R1. This robust overall performance is not driven by a narrow specialty, but by consistent, top-tier results across a wide spectrum of tasks; it secures the best scores on diverse benchmarks, including the in-domain NQ, the out-of-domain PopQA, and the adversarial Bamboogle dataset, while remaining highly competitive on all others. This consistent, high-level performance across diverse benchmarks validates that by systematically distilling, managing and utilizing, agents can develop more generalizable and powerful problem-solving strategies.

## 5 FURTHER ANALYSIS

### 5.1 ANALYSIS OF MODEL SCALES GENERALIZABILITY

---

[1]https://github.com/volcengine/verl

Table 1: Main results on QA benchmarks. The best performance in each column is set in **bold**. Our proposed model, EvolveR, is highlighted in gray.

| Methods | In domain | | Out of domain | | | | | Avg. |
|---|---|---|---|---|---|---|---|---|
| | NQ | HotpotQA | TriviaQA | PopQA | 2wiki | Musique | Bamboogle | |
| **Qwen2.5-3B** | | | | | | | | |
| Direct Inference | 0.106 | 0.149 | 0.288 | 0.108 | 0.244 | 0.020 | 0.024 | 0.134 |
| CoT | 0.023 | 0.021 | 0.032 | 0.005 | 0.021 | 0.002 | 0.000 | 0.015 |
| IRCoT | 0.111 | 0.164 | 0.312 | 0.200 | 0.171 | 0.067 | 0.240 | 0.181 |
| Search-o1 | 0.238 | 0.221 | 0.472 | 0.262 | 0.218 | 0.054 | 0.320 | 0.255 |
| RAG | 0.348 | 0.255 | 0.544 | 0.387 | 0.226 | 0.047 | 0.080 | 0.270 |
| SFT | 0.249 | 0.186 | 0.292 | 0.104 | 0.248 | 0.044 | 0.112 | 0.176 |
| R1-base | 0.226 | 0.201 | 0.455 | 0.173 | 0.268 | 0.055 | 0.224 | 0.229 |
| R1-instruct | 0.210 | 0.208 | 0.449 | 0.171 | 0.275 | 0.060 | 0.192 | 0.224 |
| Rejection Sampling | 0.294 | 0.240 | 0.488 | 0.332 | 0.233 | 0.059 | 0.210 | 0.265 |
| Search-R1-base | 0.406 | 0.284 | **0.587** | 0.435 | 0.273 | 0.049 | 0.088 | 0.303 |
| Search-R1-instruct | 0.341 | 0.324 | 0.545 | 0.378 | 0.319 | 0.103 | 0.264 | 0.325 |
| EvolveR (ours) | **0.434** | **0.373** | 0.584 | 0.434 | **0.381** | **0.137** | **0.328** | **0.382** |
| **Qwen2.5-7B** | | | | | | | | |
| Direct Inference | 0.134 | 0.183 | 0.408 | 0.140 | 0.250 | 0.031 | 0.120 | 0.181 |
| CoT | 0.048 | 0.092 | 0.185 | 0.054 | 0.111 | 0.022 | 0.232 | 0.106 |
| IRCoT | 0.224 | 0.133 | 0.478 | 0.301 | 0.149 | 0.072 | 0.224 | 0.239 |
| Search-o1 | 0.151 | 0.187 | 0.443 | 0.131 | 0.176 | 0.058 | 0.296 | 0.206 |
| RAG | 0.349 | 0.299 | 0.585 | 0.392 | 0.235 | 0.058 | 0.208 | 0.304 |
| SFT | 0.318 | 0.217 | 0.354 | 0.121 | 0.259 | 0.066 | 0.112 | 0.207 |
| R1-base | 0.297 | 0.242 | 0.539 | 0.202 | 0.273 | 0.083 | 0.296 | 0.276 |
| R1-instruct | 0.270 | 0.237 | 0.537 | 0.199 | 0.292 | 0.072 | 0.293 | 0.271 |
| Rejection Sampling | 0.360 | 0.331 | 0.592 | 0.380 | 0.296 | 0.123 | 0.355 | 0.348 |
| Search-R1-instruct | 0.393 | 0.370 | 0.610 | 0.397 | **0.414** | 0.146 | 0.368 | 0.385 |
| EvolveR (ours) | **0.462** | **0.411** | **0.620** | **0.473** | 0.395 | **0.168** | **0.392** | **0.417** |

To validate that our EvolveR framework is a generalizable paradigm rather than a method tailored to a specific model size, we evaluated its performance across a spectrum of open-source model scales. As presented in Figure 4, we applied EvolveR to Qwen2.5 models of 0.5B, 1.5B, 3B and 7B parameters. The results reveal a clear and consistent positive trend: as the parameter count of the base model increases, the performance of the EvolveR agent improves significantly on every benchmark. The average performance rises monoton-

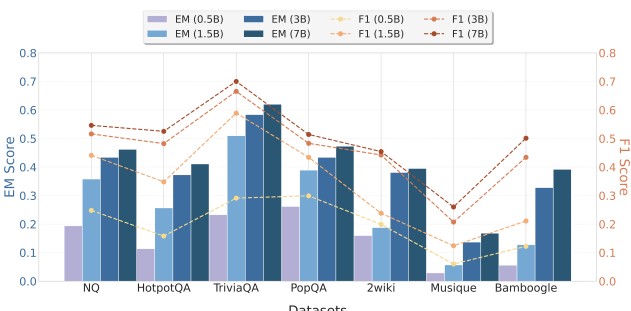

Figure 4: Performance of EvolveR across various model scales.

ically from 0.150 on the 0.5B model to 0.270 on the 1.5B model, 0.382 on the 3B model, and further to 0.417 on the 7B model. This scaling behavior demonstrates that our experience-driven lifecycle effectively harnesses the superior reasoning and instruction-following capabilities inherent in larger foundational models. It confirms that EvolveR acts as a synergistic layer on top of the base model, and suggests that its performance will continue to improve with future advancements in the open-source LLM landscape.

## 5.2 Ablation Studies: Dissecting the EvolveR Framework

### 5.2.1 Validating the Self-Distillation Mechanism

A central claim of our work is that an agent can learn effectively through self-distillation. To rigorously investigate this, we compare our standard `EvolveR (self-distill)` against a strong

Table 2: Validating the self-distillation mechanism. We compare our EvolveR, which uses its own model for distillation, against a variant that uses a larger, external model (GPT-4o-mini).

| Model Variant | In domain | | Out of domain | | | | | Avg. |
|---|---|---|---|---|---|---|---|---|
| | NQ | HotpotQA | TriviaQA | PopQA | 2wiki | Musique | Bamboogle | |
| **Qwen2.5-0.5B** | | | | | | | | |
| EvolveR (self-distill) | 0.194 | 0.114 | 0.233 | 0.262 | 0.160 | 0.029 | 0.056 | 0.150 |
| EvolveR (teacher-distill) | 0.281 ↑ | 0.193 ↑ | 0.402 ↑ | 0.363 ↑ | 0.202 ↑ | 0.033 ↑ | 0.064 ↑ | 0.220 ↑ |
| **Qwen2.5-1.5B** | | | | | | | | |
| EvolveR (self-distill) | 0.358 | 0.257 | 0.510 | 0.389 | 0.188 | 0.057 | 0.128 | 0.270 |
| EvolveR (teacher-distill) | 0.352 ↓ | 0.259 ↑ | 0.503 ↓ | 0.395 ↑ | 0.207 ↑ | 0.072 ↑ | 0.240 ↑ | 0.290 ↑ |
| **Qwen2.5-3B** | | | | | | | | |
| EvolveR (self-distill) | 0.434 | 0.373 | 0.584 | 0.434 | 0.381 | 0.137 | 0.328 | 0.382 |
| EvolveR (teacher-distill) | 0.421 ↓ | 0.372 ↓ | 0.583 ↓ | 0.359 ↓ | 0.437 ↑ | 0.127 ↓ | 0.288 ↓ | 0.370 ↓ |

alternative, `EvolveR(teacher-distill)`, which uses the powerful GPT-4o-mini as an external model for experience distillation.

The results, presented in Table 2, reveal a nuanced, scale-dependent relationship. For smaller models like the 0.5B variant, the stronger external teacher provides a clear benefit, as the base model's own distillation capabilities are limited. However, as the model scales to 3B, a reversal occurs: our `EvolveR (self-distill)` (0.382 avg.) outperforms the teacher-guided variant (0.370 avg.). This is a critical finding, suggesting that as an agent's own reasoning becomes more sophisticated, principles distilled from its own internal policy are ultimately more effective due to better "cognitive alignment". This validates self-distillation as a core, scaling strength of the EvolveR paradigm.

### 5.2.2 THE ROLE OF EXPERIENCE RETRIEVAL

To quantify the direct benefit of providing the agent with access to its distilled principles at inference time. To achieve this, we compare our full `EvolveR` model against an ablated variant, `EvolveR w/o exp-retrieve`. It is critical to note that both models undergo the identical experience-driven RL training process. The sole difference is that the `w/o exp-retrieve` variant is denied access to the experience base during evaluation.

The results in Table 3 show a stark performance degradation across all model scales when experience retrieval is disabled. For the 3B model, for instance, the average performance drops significantly from 0.382 to 0.340. This substantial gap underscores a key finding: an agent trained with our EvolveR framework, while powerful on its own, achieves its full potential only when it can access and condition on the relevant principles from its past. This demonstrates that experience retrieval is a critical and indispensable component of the EvolveR paradigm for optimal performance.

Table 3: Investigating the role of experience retrieval at inference time. The `w/o exp-retrieve` variant uses the same model but is not allowed to access the experience base during evaluation.

| Model Variant | In domain | | Out of domain | | | | | Avg. |
|---|---|---|---|---|---|---|---|---|
| | NQ | HotpotQA | TriviaQA | PopQA | 2wiki | Musique | Bamboogle | |
| **Qwen2.5-0.5B** | | | | | | | | |
| EvolveR | 0.194 | 0.114 | 0.233 | 0.262 | 0.160 | 0.029 | 0.056 | 0.150 |
| EvolveR w/o exp-retrieve | 0.085 ↓ | 0.065 ↓ | 0.137 ↓ | 0.150 ↓ | 0.082 ↓ | 0.013 ↓ | 0.016 ↓ | 0.078 ↓ |
| **Qwen2.5-1.5B** | | | | | | | | |
| EvolveR | 0.358 | 0.257 | 0.510 | 0.389 | 0.188 | 0.057 | 0.128 | 0.270 |
| EvolveR w/o exp-retrieve | 0.136 ↓ | 0.112 ↓ | 0.218 ↓ | 0.160 ↓ | 0.136 ↓ | 0.019 ↓ | 0.080 ↓ | 0.123 ↓ |
| **Qwen2.5-3B** | | | | | | | | |
| EvolveR | 0.434 | 0.373 | 0.584 | 0.434 | 0.381 | 0.137 | 0.328 | 0.382 |
| EvolveR w/o exp-retrieve | 0.405 ↓ | 0.343 ↓ | 0.569 ↓ | 0.392 ↓ | 0.334 ↓ | 0.100 ↓ | 0.240 ↓ | 0.340 ↓ |

### 5.3 EXPLORING THE INFLUENCE OF EXPERIENCE INTERNALIZATION

In our proposed framework, all retrieved information (both from the external knowledge base (<information>) and our internal experience base (<experience>)) is treated as context, with loss masked during the model update phase. A natural question arises from this design: while it is sensible to avoid learning the content of transient external documents, could the agent benefit from directly absorbing its own distilled wisdom into its parameters?

To explore this, we conducted a supplementary experiment on the Qwen2.5-3B model. We created a variant, `EvolveR w/ exp-absorb`, where we selectively unmasked the loss for the retrieved <experience> tokens, allowing the learning signal to flow through them. Our hypothesis was that this might enable the agent to internalise the strategic logic of its principles. The results, presented in Table 4, were insightful. The `EvolveR w/ exp-absorb` variant exhibited a slight performance degradation compared to our standard approach.

We posit that this degradation arises from two related factors. First, noise from irrelevant principles. In our current implementation, the agent retrieves a set of top-$k$ principles at each step, not all of which may be perfectly relevant, and directly internalising all retrieved principles without a dynamic quality filter risks updating the model with noisy or counter-productive signals. Second, a potential mismatch in the optimisation objective. Currently, unmasked experience tokens are treated identically to generated reasoning steps and optimised via GRPO's advantage-based loss, while effective internalisation might instead require treating high-quality principles as "ground truth" knowledge to be memorised. This suggests that a distinct loss formulation, such as likelihood maximisation applied specifically to the <experience> block and separate from the RL loss, might be necessary to properly absorb this wisdom without destabilising the policy.

Table 4: Ablation study on the experience internalization mechanism. EvolveR w/o exp-absorb treats principles as external context by masking gradients during backpropagation.

| Model Variant | In domain | | Out of domain | | | | | Avg. |
|---|---|---|---|---|---|---|---|---|
| | NQ | HotpotQA | TriviaQA | PopQA | 2wiki | Musique | Bamboogle | |
| EvolveR | 0.434 | 0.373 | 0.584 | 0.434 | 0.381 | 0.137 | 0.328 | 0.382 |
| EvolveR w exp-absorb | 0.433 ↓ | 0.369 ↓ | 0.583 ↓ | 0.435 ↑ | 0.376 ↓ | 0.124 ↓ | 0.280 ↓ | 0.371 ↓ |

## 6 CONCLUSION

In this work, we introduced **EvolveR**, a novel paradigm for self-evolving LLM agents centered on a complete, closed-loop experience lifecycle. Our extensive experiments demonstrate the effectiveness of this approach, showing that EvolveR consistently and significantly outperforms a wide range of strong baseline methods on a comprehensive suite of QA benchmarks. Furthermore, our detailed ablation studies rigorously validate the core tenets of our framework, confirming the significant value of the agent's self-distilled experiences and demonstrating the high efficacy of the self-distillation mechanism itself. While the quality of distilled principles is inherently tied to the base model's capabilities, pointing to promising future work, EvolveR provides a concrete blueprint for agents that learn from the consequences of their own experiences, shifting the focus from merely accessing knowledge to actively building and evolving expertise.

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

# A  APPENDIX

## A.1  USE OF LARGE LANGUAGE MODELS

The research presented in this paper, including the core ideas, experimental design, and quantitative results, is the original work of the authors. A large language model was used as a writing assistant for tasks such as polishing prose, improving clarity, and correcting grammatical errors in the manuscript. All final content was reviewed and edited by the authors to ensure it accurately reflects our research and contributions.

## A.2  EXPERIMENTAL IMPLEMENTATION DETAILS

We provide a comprehensive list of hyperparameters and implementation details used in our experiments to ensure full reproducibility.

**General Setup.**  Across all experiments, we use models from the Qwen2.5 family (Qwen et al., 2025) with their corresponding tokenizers. The maximum sequence length is set to 8192 tokens for all inputs, and the maximum response sequence length is set to 1024 tokens. The GPT-4o-mini model is used as the teacher model in the corresponding ablation study. We use BGE-M3 (Chen et al., 2024) as our embedding model.

**Cold-start Stage.**  This initial SFT stage is designed solely to teach the model the required interaction format(e.g., producing well-structured <think> and <search> actions). The model is trained for 3 epochs using the Adam optimizer, with an initial learning rate of $1 \times 10^{-4}$, a warm-up ratio of 0.1, and a batch size of 16.

**Online Interaction Phase.**  For each <search_knowledge> action, we retrieve the top-$k_d = 3$ documents from the external knowledge base, following the prior work (Jin et al., 2025). Similarly, for each <search_experience> action, we retrieve the top-$k_e = 3$ principles from the experience base $\mathcal{E}$.

**Offline Distill Phase.**  The self-distill mechanism utilizes the agent's own policy model $\pi_\theta$ to distill principles. The temperature is set to 1 during this phase. For the deduplication and integration process, we first use a semantic similarity pre-filter with a threshold of $\theta_{\text{sim}} = 0.85$ before passing candidates to the LLM-based equivalence check. The periodic principle sweep removes any principle from $\mathcal{E}$ whose `metric_score` falls below the pruning threshold of $\theta_{\text{prune}} = 0.3$.

**Reward Function Details.**  As described in the Section 3.3, the Format Reward is an average of a think score and a search score. We detail their specific calculation here. The think score $R_{\text{think}}$ is determined by a discrete mapping based on the number of <think> actions, $N_{\text{think}}$: it scales from 0.2 (for $N_{\text{think}} = 1$) to a maximum of 1.0 (for $N_{\text{think}} = 6$), and is capped at 0.5 for excessive reasoning ($N_{\text{think}} > 8$) to encourage conciseness. The search score $R_{\text{search}}$ is the sum of a diversity score and a quantity bonus. The diversity score is 0.5 if both <search_experience> and <search_knowledge> are used, 0.2 if only one type is used, and 0 otherwise. A quantity bonus of 0.1 is added for each additional search action beyond the first, up to a maximum bonus of 0.5 (for a total of 6 searches).

**Policy Optimization.**  The composite reward function is weighted with $w_o = 1.0$ for the outcome reward and $w_f = 0.1$ for the format reward. For the GRPO objective function (Equation 5), the clipping parameter is set to $\epsilon = 0.2$ and the KL-divergence coefficient is $\beta = 0.001$. During the training procedure, we adopt vLLM to accelerate LLM rollouts. The tensor parallel size is set to 1, and the GPU memory utilization ratio is set at 0.6. For rollout sampling, we use a temperature of 1.0 and a top-p value of 0.95.

**Computational Cost Analysis.**  We provide detailed computational resources required for training and experience retrieval latency to demonstrate the efficiency of EvolveR.

- **Training Cost:** The full training lifecycle for the Qwen2.5-3B model, which includes the cold-start SFT phase and the subsequent RL policy evolution via GRPO, requires approximately **39.4 hours** on a server equipped with 8 NVIDIA A100 (80GB) GPUs.

- **Experience Retrieval Latency:** A key concern for retrieval-augmented systems is the added latency during inference. We implement the Experience Base using Milvus for efficient vector similarity search. Our empirical measurements show that even with an experience base containing approximately **14,000 principles**, the latency for retrieving the top-3 relevant principles is approximately **0.06 seconds**. That is imperceptible compared to LLM generation time, ensuring that EvolveR maintains high throughput during deployment.

**SFT-only Baseline Details.** We used the exact same set of successful trajectories generated during the Online Interaction stage. Both <experience> and <information> tokens were masked during training. We utilised the LLaMAFactory Zheng et al. (2024) to fine-tune the Qwen2.5-3B model using LoRA (Low-Rank Adaptation). The LoRA rank is set to 8, the learning rate is $1 \times 10^{-4}$, and the model is trained for 5 epochs with a batch size of 1. The warmup ratio is 0.1, the maximum sequence length is 4096 tokens, gradient accumulation steps are set to 8, and the learning rate scheduler is `Cosine`.

### A.3 ADDITIONAL EXPERIMENTAL ANALYSIS

#### A.3.1 NECESSITY OF THE RL (GRPO) STAGE

To assess the necessity of the Reinforcement Learning stage in `EvolveR`, we conduct an ablation experiment on the Qwen2.5-3B model. Specifically, we reuse the exact same set of successful trajectories collected during online interaction, but train the policy via standard SFT rather than GRPO. The full SFT hyperparameter configurations are provided in Appendix A.2.

As shown in Table 5, the RL-based variant substantially outperforms the SFT-only version on the 3B model, achieving a 7% relative improvement. This result highlights the fundamental limitation of SFT: it merely encourages the model to reproduce surface-level action sequences from successful trajectories, without understanding the underlying utility or expected reward of actions such as <search_experience>. In contrast, our RL approach (GRPO) leverages both successful and failed trajectories, enabling the agent to learn *what to do* from positive rollouts and *what to avoid* from negative ones, which is essential for developing robust retrieval and reasoning strategies.

Table 5: Comparison between SFT-only and RL/GRPO training in EvolveR on the Qwen2.5-3B model.

| Model Variant | In domain | | Out of domain | | | | | Avg. |
|---|---|---|---|---|---|---|---|---|
| | NQ | HotpotQA | TriviaQA | PopQA | 2wiki | Musique | Bamboogle | |
| EvolveR(SFT) | 0.415 | 0.357 | 0.584 | 0.419 | 0.366 | 0.106 | 0.248 | 0.357 |
| EvolveR(RL) | **0.434** | **0.373** | **0.584** | **0.434** | **0.381** | **0.137** | **0.328** | **0.382** |

#### A.3.2 HYPERPARAMETER SENSITIVITY ANALYSIS

To assess the robustness of EvolveR and the impact of the dynamic experience curation mechanism, we conducted a sensitivity analysis on the pruning threshold $\theta_{\text{prune}}$. This parameter dictates the minimum `metric_score` required for a principle to be retained in the Experience Base after each cleaning. We evaluated the performance of the EvolveR-1.5B model across a wide range of thresholds: $\theta_{\text{prune}} \in \{0.1, 0.3, 0.7, 0.9\}$.

The results confirm that performance remains robust across different thresholds. Our default setting ($\theta_{prune} = 0.3$) effectively filters out low-quality principles to prevent the database from growing indefinitely.

Table 6: Sensitivity analysis of the pruning threshold $\theta_{\text{prune}}$ on the Qwen2.5-1.5B model.

| $\theta_{\text{prune}}$ | In domain | | Out of domain | | | | | Avg. |
|---|---|---|---|---|---|---|---|---|
| | NQ | HotpotQA | TriviaQA | PopQA | 2wiki | Musique | Bamboogle | |
| 0.1 (loose) | 0.336 | 0.244 | 0.496 | 0.387 | **0.207** | 0.052 | 0.160 | 0.269 |
| 0.3 (Default) | **0.358** | **0.257** | **0.510** | **0.389** | 0.188 | 0.057 | 0.128 | 0.270 |
| 0.7 (strict) | 0.276 | 0.253 | 0.498 | 0.379 | 0.196 | 0.060 | **0.192** | 0.265 |
| 0.9 (very strict) | 0.323 | **0.257** | **0.510** | 0.385 | 0.204 | **0.061** | **0.192** | **0.276** |

### A.3.3 LONGITUDINAL ANALYSIS OF LEARNING DYNAMICS

To provide a deeper understanding of how the EvolveR agent improves over time, we present a longitudinal analysis of its behavior during the RL training process. We focus on two key aspects: the evolution of action frequencies and the improvement in the quality of distilled principles.

**Evolution of Action Frequencies.** We tracked the average number of `<think>`, `<search_knowledge>` and `<search_experience>` actions per trajectory across training steps. As shown in Figure 5, the agent's behavior exhibits distinct phases of optimization:

- **Early Phase (Interval 1):** Thanks to the cold-start SFT, the agent begins with reasonable tool usage capabilities.

- **Optimization Phase (Intervals 2-3):** As RL training progresses, we observe a clear upward trend in the frequency of `<think>` and `<search_knowledge>`. This indicates that the agent is learning to engage in deeper reasoning and more extensive external information gathering to solve complex tasks.

- **Convergence Phase (Interval 4):** Crucially, the action counts do not increase indefinitely. They converge to a stable, efficient range (approx. 2 experience searches, 4 knowledge searches, and 6 reasoning steps).

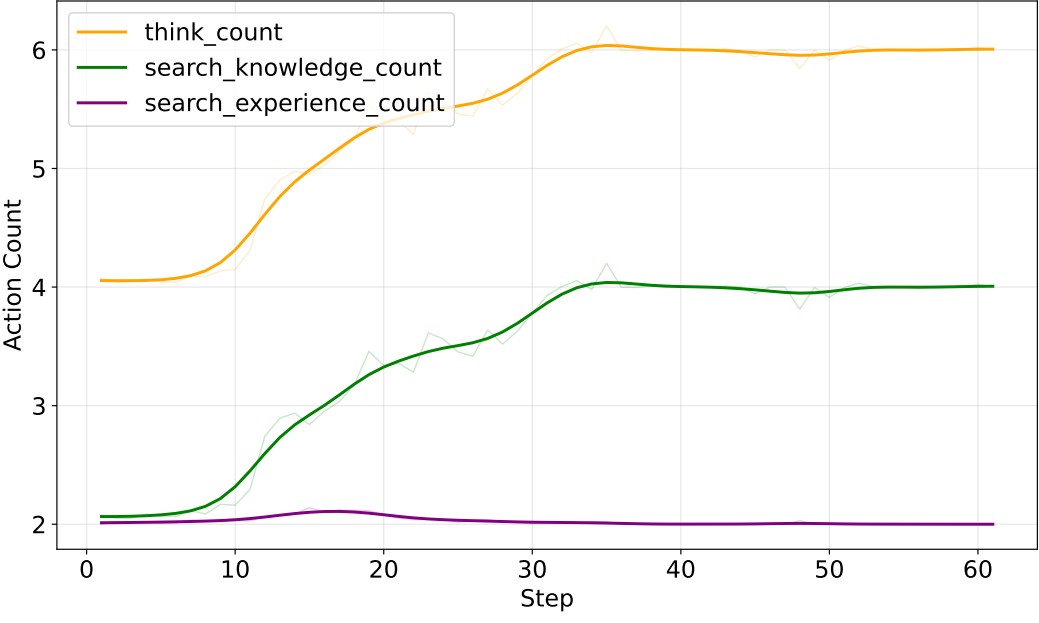

Figure 5: Evolution of average action counts per trajectory during RL training.

**Intrinsic Improvement in Principle Distillation.** To verify that the agent is indeed improving its ability to distill high-quality principles (rather than just filtering out bad ones), we analyzed the quality of principles grouped by their distilled time. We divided the training process into four

intervals and calculated the average `metric_score` of the principles generated in each interval that remain in the Experience Base.

Table 7: Quality of principles by creation phase.

| Creation Phase | Avg. Metric Score | Avg. Usage Count |
|---|---|---|
| Interval 1 | 0.462 | 8.22 |
| Interval 2 | 0.464 | 24.10 |
| Interval 3 | 0.487 | 17.82 |
| Interval 4 | **0.500** | 12.36 |

Table 7 presents the evolution of principle quality across different training stages. Principles from Interval 1 have undergone the longest duration of dynamic pruning, representing a highly filtered subset. Conversely, principles from Interval 4 are relatively new and have been subject to less filtering. We observe that the principles generated in the final stage achieve a higher average metric score (0.500) compared to those from the earliest stage (0.462). Supported by substantial usage counts ($> 12$) that ensure statistical stability, this trend indicates that the agent's intrinsic ability to distill high-quality strategies improves over time.

## A.4 Prompt Details

### A.4.1 Cold Start Prompt

The Prompt in Table A.4.4 is used during the cold-start stage to generate the initial trajectories for SFT. This prompt guides a powerful LLM (we used GPT-4o) to act as an expert problem-solver, producing a small dataset with the right format trajectories to cold start. It is not used to teach the cold-start model knowledge. Once this brief cold start phase is completed, the dependency on the external model is removed. All subsequent processes (including reasoning with principles, principle distillation, and policy updates) are driven solely by the agent itself.

### A.4.2 System Prompt

The Prompt in Table A.4.4 is the system prompt used by the EvolveR agent during the online interaction phase. It defines the agent's core identity, its available actions (<think>, <search_knowledge>, <search_experience>, <answer>), and the overall format for its reasoning process.

### A.4.3 Distill Principle Prompt

The Prompt in Table A.4.4 and Table A.4.4 are used during the offline experience self-distillation phase to enable the agent's self-distillation mechanism. Based on the outcome of a trajectory, one of two distinct prompts is used to guide the agent's own model ($\pi_\theta$) to distill a principle. The first Prompt is for successful trajectories, focusing on extracting a guiding principle. The second is for failed trajectories, aimed at formulating a cautionary principle.

### A.4.4 Judge Same Principle Prompt

The Prompt in Table A.4.4 is a crucial component of the deduplication and integration process within the offline experience self-distillation. It tasks the agent's own model ($\pi_\theta$) with acting as a semantic judge. Given two principles (a newly distilled candidate and a retrieved existing similar one), the model is asked to determine if they are semantically equivalent. The binary "yes/no" output of this Prompt is used to decide whether to merge a new experience or create a new principle.

## A.5 Case Study of EvolveR

### A.5.1 Qualitative Analysis of Distilled Principles

To provide concrete insight into the nature of the wisdom distilled by EvolveR, we present a qualitative comparison between a low-scoring (eliminated) principle and a high-scoring (retained) principle from our Experience Base.

Table 8: System prompt for LLM agents

Answer the given question.

You must conduct reasoning inside <think> and </think> first every time you get new information or get new experience principles.
After reasoning, you can search for past experiences by <search_experience> query </search_experience> to get relevant past experience principles (may be guilding or warning principles) and it will return the top searched results between <experience> and </experience>.
You can use these principles which you think is helpful to help you answer the question.

If you find you lack some knowledge, you can call a search engine by <search_knowledge> query </search_knowledge> and it will return the top searched results between <information> and </information>.
You can search knowledge and experience as many times as your want.

If you find no further external knowledge needed, you can directly provide the answer inside <answer> and </answer>, without detailed illustrations.
For example, <answer> Beijing </answer>

**User: {question}**

As shown in Table 13, the distinction is clear. The high-scoring principle provides a precise heuristic for a specific reasoning scenario (character verification), guiding the model on *what to avoid* and *how to verify*. In contrast, the low-scoring principle offers abstract advice about the reasoning process itself ("avoid redundancy"), which lacks the specificity required to guide the agent's actual actions. This comparison validates that our dynamic scoring mechanism successfully filters for practical utility, retaining wisdom that is grounded and executable.

### A.5.2 ROLLOUT CASE

We provide a detail rollout case of EvolveR in Table A.5

### A.6 LIMITATION AND BROADER IMPACT

We acknowledge several limitations and broader implications of our work. The efficacy of our self-distillation mechanism is inherently bounded by the capabilities of the agent's own model; a less capable model may struggle to distill high-quality principles, thus limiting its evolutionary ceiling. Further research across a broader range of tasks, such as embodied interaction or creative generation, is necessary to fully delineate the boundaries and applicability of the EvolveR paradigm. While our curation mechanisms mitigate experience base growth, ensuring computational efficiency for truly lifelong learning agents also remains an open challenge. Looking forward, the broader impact of this paradigm is significant. On the one hand, EvolveR represents a crucial step towards more autonomous and personalized agents. The explicit nature of its distilled principles also offers a promising avenue for improving interpretability and steerability. On the other hand, this autonomy raises critical safety considerations. An agent that evolves its own principles could develop undesirable strategies if not guided by a robust, value-aligned reward function, necessitating further research into alignment techniques for such self-evolving systems.

## B   ETHICS STATEMENT

Our work on self-evolving agents is a foundational research exploration in controlled, simulated environments. We acknowledge that the deployment of such autonomous learning systems in the real world would raise significant safety and alignment challenges, as an agent could potentially develop undesirable strategies. We believe the inherent interpretability of EvolveR's distilled, human-readable principles offers a promising direction for mitigating these risks through human oversight, and we advocate for further research in alignment with such evolving systems before any high-stakes application is considered. This work does not involve any personally identifiable information or sensitive data.

Table 9: Prompt for cold start.

You are a top-notch intelligent reasoning expert, adept at restoring solution paths from given answers and documents in reverse. Your task is to simulate a full reasoning trajectory for answering the question below, based on the provided documents and answer. You must reason step-by-step as if you do not yet know the final answer, even though it is given for supervision.

In <think> blocks, do not reference or confirm the final answer directly. Instead, reason like a human—understand the task, recall prior knowledge, evaluate the need for experience or external information, and gradually infer the answer.

The reasoning trajectory must follow the **exact format below**. If the retrieved **experience alone is sufficient to answer the question**, you may skip the <search_knowledge> and <information> steps.

**Output Format:**

<think> ... </think>
<search_experience>
- Retrieve 2–3 relevant abstract experience principles, using structured triple format.
- For each principle, add a short description of its purpose.
</search_experience>
<think> Explain what you plan to do after retrieving experience. Decide whether you still need to retrieve knowledge. </think>

[IF experience is enough:]
<think>
- List the principles you are applying, include their triple form and description.
- Explain briefly how each principle contributes to your reasoning.
- Continue with reasoning based on these principles and conclude with your final judgment.
</think>
<answer>...</answer>

[ELSE:]
<search_knowledge>
- Generate one or more natural language search queries that would help retrieve the provided documents.
</search_knowledge>
<information>
{relevant_document}
</information>
<think> Reflect on retrieved information. </think>
<think>
- List the principles you are applying, include their triple form and description.
- Explain how each principle guides the reasoning process using the retrieved information.
- Summarize your reasoning path and justify the answer.
</think>
<Answer>...</Answer>

**Inputs:**

**Query:** {query}
**Relevant Documents:** {relevant_document}
**Answer:** {answer}

Please begin generating the reasoning trajectory.

## C  REPRODUCIBILITY STATEMENT

To ensure full reproducibility, we will release all code for the EvolveR framework, training scripts and evaluation scripts on GitHub upon publication. Our experiments are conducted on publicly available benchmarks (e.g., NQ, HotpotQA) using open-source Qwen2.5 models. All critical hyper-parameters, including those for the SFT cold-start, GRPO optimization and the experience curation

Table 10: Prompt for summarizing a successful interaction trajectory.

You are an expert in analyzing interaction logs to distill generalizable wisdom.
Analyze the following successful interaction trajectory. Your goal is to extract a "Guiding Principle" from it.

A "Guiding Principle" has two parts:
1. A concise, one-sentence natural language description. This is the core advice.
2. A structured representation of the key steps or logic, as a list of simple (subject, predicate, object) triplets.

**[Trajectory Log]**:
{{trajectory_log}}
Final Outcome: SUCCESS

**Your Task:**
Based on the trajectory, generate the Guiding Principle.
First, on a new line, write {DESCRIPTION_PART_SEPARATOR}.
Then, write the one-sentence description of the pitfall.
Then, on a new line, write {STRUCTURED_PART_SEPARATOR}.
Finally, provide the structured triplets describing the failure pattern in a valid JSON list format.

**[Example]**:
{DESCRIPTION_PART_SEPARATOR}
When a file download fails with a 404 error, do not immediately retry the download; instead, verify the source URL's validity first.
{STRUCTURED_PART_SEPARATOR}

```
[
    (file download, results_in, 404 error),
    (immediate_retry, is, ineffective),
    (correct_action, is, verify URL)
]
```

**[Output]**:

pipeline, are detailed in Appendix A.2. The primary computational requirement is a server with 8 A100 GPUs for training the 3B model.

Table 11: Prompt for summarizing a failed interaction trajectory.

You are an expert in analyzing interaction logs to find the root cause of failures.
Analyze the following failed interaction trajectory. Your goal is to extract a "Cautionary Principle" from it.

A "Cautionary Principle" has two parts:
1. A concise, one-sentence description of the key mistake to avoid and under what circumstances.
2. A structured representation of the failure pattern, as a list of simple (subject, predicate, object) triplets.

**[Trajectory Log]**:
{{trajectory_log}}
Final Outcome: FAILURE

**Your Task:**
Based on the trajectory, generate the Cautionary Principle.
First, on a new line, write {DESCRIPTION_PART_SEPARATOR}.
Then, write the one-sentence description of the pitfall.
Then, on a new line, write {STRUCTURED_PART_SEPARATOR}.
Finally, provide the structured triplets describing the failure pattern in a valid JSON list format.

**[Example]**:
{DESCRIPTION_PART_SEPARATOR}
When a file download fails with a 404 error, do not immediately retry the download; instead, verify the source URL's validity first.
{STRUCTURED_PART_SEPARATOR}

```
[
    (file download, results_in, 404 error),
    (immediate_retry, is, ineffective),
    (correct_action, is, verify URL)
]
```

**[Output]**:

Table 12: Prompt for Principle Similarity Analysis.

You are a semantic analysis expert. Determine if two principles describe the same core idea, even if they use different words.
Principle A: "{summary}"
Principle B: "{existing_principle_description}"

Do Principle A and Principle B describe the same essential advice or warning?
Please answer with only "Yes" or "No".

Table 13: Qualitative comparison of principles.

| Type | Score | Principle Content | Analysis |
|---|---|---|---|
| **Low Score** *(Eliminated)* | 0.25 | *"When using multiple principles, ensure there is no redundancy or unnecessary overlapping, leading to confusion about the principle applicability and efficiency."* | **Vague:** This is a generic meta-statement about the process, not a specific strategy for solving the task. It offers no actionable guidance. |
| **High Score** *(Retained)* | 0.80 | *"When determining character relationships in a series, avoid assuming actors based solely on character recognition; confirm with verified role descriptions."* | **Specific & Actionable:** This principle identifies a specific pitfall (hallucinating actors) and provides a concrete correction strategy. |

Table 14: A rollout case of EvolveR

**System**: System_Prompt

**Question**: who was the pinkerton detective agency's first female detective?

**Ground Truth**: Kate Warne

**EvolveR**: <think>I need to determine who was the first female detective employed by the Pinkerton Detective Agency. This requires identifying the historical context and any notable female figures associated with this agency. Based on prior reasoning experience, I anticipate some potentially useful principles:- Principle 1: (Historical Agency, First Female Employee, Female Detective) - This principle helps identify the first female to hold a specific role within a historical organization. - Principle 2: (Organization, Historical Role, Notable Individual) - This principle assists in linking an organization to a notable individual who played a significant role in its history.</think> <search_experience>- Principle 1: (Historical Agency, First Female Employee, Female Detective) - This principle helps identify the first female to hold a specific role within a historical organization. - Principle 2: (Organization, Historical Role, Notable Individual) - This principle assists in linking an organization to a notable individual who played a significant role in its history.</search_experience>

**Env**: <experience>[Principle 0], type: guiding, metric score: 0.67, description: To identify the first woman to hold a specific significant position within an organization or entity, connect the historical narrative context and mention which event or timeline may signify the position in question...<experience>

**EvolveR**: <think>Based on the experience principles, I plan to apply the principles related to identifying the first female employee in a historical organization and linking this to notable figures. I will also consider using historical documents or specific timelines to confirm the answer.</think> <search_knowledge>- Query: who was the first female detective employed by the Pinkerton Detective Agency?</search_knowledge>

**Env**: <information>Doc 1(Title: "Kate Warne")... Doc 2(Title: "Private investigator") ... Doc 3(Title: "Kate Warne")...</information>

**EvolveR**: <think>Based on the retrieved documents, I need to determine who was the first female detective employed by the Pinkerton Detective Agency. I will apply the principles to connect the historical context and notable figures mentioned. - Principle 1: (Historical Agency, First Female Employee, Female Detective) - This principle helps identify the first female detective within the Pinkerton Detective Agency. - Principle 2: (Organization, Historical Role, Notable Individual) - This principle links the Pinkerton Detective Agency to Kate Warne as the first female detective.</think> <search_knowledge>- Query: who was the first female detective employed by the Pinkerton Detective Agency?</search_knowledge>

**Env**: <information>Doc 1(Title: "Kate Warne") ... Doc 2(Title: "Private investigator") ... Doc 3(Title: "Kate Warne") ...</information>

**EvolveR**: <think>Based on the retrieved documents, Kate Warne is identified as the first female detective employed by the Pinkerton Detective Agency. This aligns with the historical context and the principles applied. - Principle 1: (Historical Agency, First Female Employee, Female Detective) - This principle confirms Kate Warne as the first female detective within the Pinkerton Detective Agency. - Principle 2: (Organization, Historical Role, Notable Individual) - This principle links Kate Warne to her role as the first female detective in the agency's history.</think> <search_knowledge>- Query: who was the first female detective employed by the Pinkerton Detective Agency?</search_knowledge>

**Env**: <information>Doc 1(Title: "Kate Warne") ... Doc 2(Title: "Private investigator") ... Doc 3(Title: "Kate Warne")...</information>

**EvolveR**: <think>Based on the retrieved documents, Kate Warne is confirmed as the first female detective employed by the Pinkerton Detective Agency. The information aligns with both historical documentation and the application of principles regarding notable figures in organizational history. - Principle 1: (Historical Agency, First Female Employee, Female Detective) - This principle confirms Kate Warne as the first female detective within the Pinkerton Detective Agency. - Principle 2: (Organization, Historical Role, Notable Individual) - This principle links Kate Warne to her role as the first female detective in the agency's history.</think>

<answer>Kate Warne</answer>

