# OpenReview forum: "EvolveR: Self-Evolving LLM Agents through an Experience-Driven Lifecycle"
_ICLR.cc/2026/Conference — Submitted to ICLR 2026_

### Official Review · Reviewer_J5YF · 2025-10-26

**Soundness:** 3
**Presentation:** 3
**Contribution:** 2
**Rating:** 4
**Confidence:** 4

**Summary:**

The paper introduces EvolveR, a self-evolving LLM agent framework that learns from its own interactions through a closed-loop experience lifecycle. Offline, the agent self-distills past trajectories into abstract, reusable strategic principles and maintains a curated experience base via semantic deduplication, integration, and dynamic scoring. Online, it retrieves these principles to guide multi-turn reasoning and tool use, generating higher-quality trajectories that feed back into learning. The loop is closed with reinforcement learning (GRPO), optimizing the policy on experience-guided trajectories to internalize effective strategy use. Across seven QA benchmarks and multiple Qwen2.5 model scales, EvolveR consistently outperforms strong baselines, with the 3B model achieving the best results; ablations show experience retrieval is indispensable and that self-distillation surpasses distillation by a stronger external teacher at larger scales, supporting the importance of cognitive alignment.

**Strengths:**

1. The paper is in general well-written.
2. The results are strong and surpass search-R1 by 5.7% on average.

**Weaknesses:**

1. This paper claims to propose a novel framework to leverage distilled experiences in online interaction. I feel this core idea lacks novelty, as it is similar to existing work that learns from experience (https://arxiv.org/pdf/2504.06821?, https://arxiv.org/pdf/2402.12317) and the data synthesis from existing experiences (https://arxiv.org/pdf/2501.10893?)
2. The evaluated datasets are limited to QA tasks. It would be better if the authors could extend to more general scenarios.
3. It is unclear that whether the generated data would be effective for larger models.

**Questions:**

Have you compared the difference between SFT and RL? What happens if you use SFT instead of RL.

---

> ### Author Response · Authors · 2025-11-23
> **Response [1/2]**
>
> We sincerely thank the reviewer for the positive assessment of our writing and the recognition of our empirical results. We appreciate the critical and insightful comments regarding novelty, scope and learning paradigms. We address these points in detail below.
>
> # Response to Weakness 1: Novelty and Distinction from Existing Works
> We thank the reviewer for bringing these related works to our attention. We have carefully analyzed them and clarify the fundamental distinctions of EvolveR below:
>
> - ASI [1] focuses on encapsulating primitive actions into higher-level skills specifically for web navigation efficiency. Its contribution is primarily in action space optimization rather than general strategy learning.
> - Learn-by-Interact [2] addresses data scarcity by synthesizing trajectories from documents to perform fine-tuning. It is essentially a data augmentation technique for cold-start, lacking a dynamic, self-improving lifecycle.
> - EVOR [3] targets better code generation by optimizing retrieval queries. It focuses on tool-level improvements for a specific domain rather than agent-level policy evolution.
>
> **The Unique Contribution of EvolveR:** Unlike these works which focus on specific action optimizations, data synthesis or tool-level retrieval, EvolveR proposes a Experience-Driven Self-Evolution Paradigm.
> 1. **Systematic Lifecycle:** We integrate offline self-distillation, dynamic principle curation (distill, deduplicate, update and etc.), and online agent policy update into a closed loop.
> 2. **Policy Evolution via RL:** Crucially, we go beyond simply providing experience as context (as in RAG or in-context learning). We employ RL (GRPO) to optimize the agent's policy, teaching it how to effectively utilize these distilled principles to guide multi-step reasoning and maximize task success. This active learning process distinctively separates EvolveR from methods that rely solely on static retrieval or imitation.
>
> [1] Wang Z Z, Gandhi A, Neubig G, et al. Inducing programmatic skills for agentic tasks[J]. arXiv preprint arXiv:2504.06821, 2025.
>
> [2] Su H, Sun R, Yoon J, et al. Learn-by-interact: A data-centric framework for self-adaptive agents in realistic environments[J]. arXiv preprint arXiv:2501.10893, 2025.
>
> [3] Su H, Jiang S, Lai Y, et al. Evor: Evolving retrieval for code generation[C]//Findings of the Association for Computational Linguistics: EMNLP 2024. 2024: 2538-2554.
>
> # Response to Weakness 2: Generality Beyond QA Tasks
> We acknowledge the reviewer's valid point regarding the evaluation scope. We selected these specific QA benchmarks because they require the agent to perform multi-step document search and complex reasoning to generate answers. These tasks serve as rigorous proxies for evaluating an agent's ability to plan strategically and process information. We agree that the EvolveR framework is inherently task-agnostic, and we are committed to extending it to broader domains such as coding and embodied interaction in future work to further validate its generalizability.

---

> ### Author Response · Authors · 2025-11-23
> **Response [2/2]**
>
> # Response to Weakness 3: Effectiveness on Larger Models
> This is an excellent suggestion. To verify the scalability of our approach, we have conducted additional experiments using the \textbf{Qwen2.5-7B} model during the rebuttal period.
>
>
> **Table: Scaling Performance of EvolveR**
> | Method | Avg. Score |
> | :--- | :---: |
> | **EvolveR-instruct (0.5B)** | **0.150** |
> | **EvolveR-instruct (1.5B)** | **0.270** |
> | **EvolveR-instruct (3B)** | **0.382** |
> | **EvolveR-instruct (7B)** | **0.417** |
>
> As shown in the table above, EvolveR (7B) achieves an average score of **0.417**,  outperforming EvolveR (3B). This confirms that our experience-driven lifecycle scales effectively: larger models, with their stronger reasoning capabilities, are even better at distilling high-quality principles and utilizing them to evolve their policies.
>
>
> # Response to Question 1: SFT vs. RL
>
> We thank the reviewer for suggesting this critical comparison. To validate the necessity of our Reinforcement Learning (GRPO) stage, we conducted an ablation study where we replaced the RL process with Supervised Fine-Tuning (SFT).
> Setup: We used the exact same set of successful trajectories generated during the Online Interaction stage. Instead of using GRPO, we fine-tuned the agent on these positive trajectories using standard SFT.
>
> **Table: Ablation Study: SFT vs. RL (Qwen2.5-3B)**
>
> | Method | Avg. Score |
> | :--- | :---: |
> | EvolveR (SFT) | 0.357 |
> | EvolveR (RL/GRPO) | **0.382** |
>
>
> The results show that EvolveR (RL) significantly outperforms EvolveR (SFT) by 7\% relative improvement. This validates our design choice:
> 1. Learning vs. Imitation: SFT merely teaches the model to imitate the surface-level actions of successful trajectories. It does not fundamentally understand the utility or expected reward behind specific actions like <search_experience>.
> 2. Contrastive Learning: Our RL approach (GRPO) leverages both successful and failed trajectories (via group relative rewards) to teach the agent **what to do** and **what to avoid**.

---

> ### Comment · Reviewer_J5YF · 2025-11-25
> **Thank you for the response**
>
> Thanks a lot for the additional experiments and results. I understand the difference between EvolveR and existing works: this work uses experiences for RL, while previous efforts use it for SFT or in-context learning. I acknowledge this contribution and would like to leave the decision to AC.

---

> > ### Author Response · Authors · 2025-11-26
> >
> > We sincerely thank the reviewer for the response and for explicitly acknowledging the core contribution of EvolveR: **leveraging experience for Policy Evolution via RL**, which distinctively differs from previous SFT paradigms. We are glad that our additional ablation study (showing a **7\% relative improvement** of RL over SFT) and the 7B model experiments helped clarify the effectiveness and scalability of this approach.
> >
> > We will ensure that the discussion regarding these distinctions and the new experimental results are incorporated into the camera-ready version. We truly appreciate your time and constructive feedback, which have significantly improved the quality of our manuscript.

---

### Official Review · Reviewer_bDce · 2025-10-27

**Soundness:** 2
**Presentation:** 4
**Contribution:** 4
**Rating:** 4
**Confidence:** 3

**Summary:**

This study proposes EvolveR, a framework that enables large language model (LLM) agents to self-evolve through their own interactive experiences.
Unlike previous approaches that rely on teacher models for knowledge distillation or external memory for storing prompts, this work introduces a human-inspired learning lifecycle consisting of two key stages.

The offline experience self-distillation phase allows the agent to summarize abstract strategic principles from its own interaction trajectories, performing semantic and quality evaluations to form an updatable experience base.
The online interaction phase enables the agent to retrieve its past strategic principles as guidance to enhance reasoning, while also generating new trajectories to further refine the experience base.

Experiments on several commonly used QA benchmarks demonstrate that EvolveR exhibits solid performance, and the ablation studies highlight the crucial role of the experience self-distillation mechanism.

**Strengths:**

1. The paper proposes a novel learning framework that models a human-inspired self-reflective learning mechanism for LLM-based agents. This framework enables agents to learn from their own experiences and enhance reasoning capabilities. The authors also include ablation studies demonstrating the effectiveness of the self-distillation component, and the framework shows competitive performance across multiple QA benchmarks.

2. The design of a continuously updatable experience database and a corresponding reward mechanism tailored to this reflective reasoning paradigm represents a useful contribution that could inspire future research and applications in related domains.

3. The writing is clear and fluent, and the figures and diagrams are well-organized, making the overall methodology easy to follow.

**Weaknesses:**

1. In the main experiments, the authors compare EvolveR primarily against Search-R1. However, other concurrent approaches mentioned in the related work—specifically Auto-Refine and O2-Searcher—are not directly compared in the experiments.
Based on available preprints from May 2025 using the same Qwen2.5-3B backbone, those methods achieved average Exact Match scores of 0.405 and 0.391, both higher than the 0.382 reported for EvolveR.
The paper only briefly explains this omission by stating:

“However, these systems either store raw, unstructured data or rely on memory mechanisms that are not designed for the systematic, long-term distillation and refinement of abstract strategic knowledge...”
While this provides conceptual differentiation, the lack of experimental comparison weakens the empirical persuasiveness of the work.

2. The paper does not provide a detailed discussion of the computational cost or resource consumption associated with maintaining and updating the strategy trajectory database. As the experience base grows, this could introduce scalability concerns and potential efficiency challenges for lifelong learning, even with pruning mechanisms.

3. A minor limitation is the absence of experiments on other LLM architectures such as LLaMA. Without such comparisons, it remains unclear whether the framework is specific to Qwen2.5 or generalizable to other base models.

**Questions:**

1. Could the authors clarify how EvolveR is positioned relative to Auto-Refine and O2-Searcher?
Are there specific scenarios where EvolveR performs better or offers complementary advantages?
Providing either empirical evidence or a clearer theoretical justification would help strengthen the experimental credibility.

2. Is there any analysis of computational efficiency, including training time or the impact of the cosine similarity threshold (θ_sim) used to decide when to add or merge trajectory principles?

3. Have the authors attempted experiments with larger-scale models (e.g., 7B or above)?
If not, could they clarify whether this was due to computational limitations or other design considerations?

---

> ### Author Response · Authors · 2025-11-23
> **Response [1/2]**
>
> We sincerely thank the reviewer for the critical and insightful comments. We address these points below.
>
> # Response to Weakness 1 & Question 1: Comparison with Auto-Refine & O2-Searcher
>
> We thank the reviewer for highlighting these important concurrent works. We acknowledge that on the Qwen2.5-3B scale, EvolveR's average score  is slightly lower than Auto-Refine and O2-Searcher. However, EvolveR occupies a distinct and complementary research area relative to these approaches. The performance differences stem primarily from **divergent experimental setups and architectural goals**:
>
> ### 1. Distinction from O2-Searcher (Data-Centric vs. Mechanism-Centric):
> O2-Searcher achieves its performance by leveraging a **significantly broader mixture of training data**, combining both open-ended and closed-ended QA datasets. In contrast, EvolveR strictly follows the standard Search-R1 setting (training only on NQ & HotpotQA) to control variables. Our goal was not to maximize scores via data scaling, but to validate the **efficacy of the experience-driven lifecycle itself**. EvolveR demonstrates that a self-evolving loop can yield substantial gains even with constrained in-domain data.
>
> ### 2. Distinction from Auto-Refine (Inference Module vs. Policy Evolution):
> Auto-Refine introduces an auxiliary "refinement module" to post-process retrieved information. This is a **"plug-in" component** that enhances inference. EvolveR, conversely, focuses on **Policy Evolution**: using RL to fundamentally update the agent's internal parameters to evolve strategic behaviors. These approaches are complementary; one could theoretically integrate Auto-Refine's module into EvolveR's lifecycle.
>
> While Auto-Refine and O2-Searcher set impressive baselines through data scaling and inference modules, EvolveR contributes a unique **Self-Evolution Paradigm**. Our work validates that an agent can autonomously distill and undate policy from its own history, a foundational capability for lifelong learning agents that is distinct from the contributions of the two works above.
>
>
> # Response to Weakness 2 & Question 2: Computational Cost & Efficiency
> We thank the reviewer for this practical question regarding resource consumption. We provide a detailed efficiency analysis below:
> 1.  **Training Cost:** The full training lifecycle (including the SFT cold-start and RL via GRPO) for the Qwen2.5-3B model takes approximately 39.4 hours on a server with 8 A100 GPUs.
> 2.  **Experience Retrieval Latency:** The overhead introduced by the experience mechanism during inference is minimal. Our tests show that even with an experience base containing approximately **14,000 principles**, retrieving the top-3 relevant principles using vector indexing takes only **~0.06 seconds**, which is negligible compared to the model's generation time.
> 3.  **Database Maintenance Analysis:** To address concerns about the long-term maintenance of the experience base, we conducted a new sensitivity analysis on the pruning threshold $\theta_{prune}$ using the Qwen2.5-1.5B model.
>
> **Table: Sensitivity Analysis of $\theta_{prune}$**
>
> | $\theta_{prune}$ | 0.1 (Loose) | 0.3 (Default) | 0.7 (Strict) | 0.9 (Very Strict) |
> | :--- | :---: | :---: | :---: | :---: |
> | **Avg. Score** | 0.269 | 0.270 | 0.265 | 0.276 |
>
> The results confirm that performance remains robust across different thresholds. Our default setting ($\theta_{prune}=0.3$) effectively filters out low-quality principles to prevent the database from growing indefinitely. We will include these detailed results in the Appendix of the revised manuscript.

---

> ### Author Response · Authors · 2025-11-23
> **Response [2/2]**
>
> # Response to Weakness 3: Generalization across Architectures
> We acknowledge that we focused on the Qwen2.5 family. Given the constraints on computational resources during the rebuttal period, we prioritized evaluating scalability across model sizes (0.5B, 1.5B, 3B, and now 7B) within a single high-quality family to control variables.
> The consistent performance gains across this wide range of sizes (from 0.5B to 7B) strongly suggest that the EvolveR mechanism is not an artifact of a specific model checkpoint, but a generalizable paradigm that scales with the underlying reasoning capability of the LLM. We are confident this extends to other architectures like LLaMA and will include these experiments in the camera-ready version.
>
>
> # Response to Question 3: Experiments on Larger Models (7B)
>
> We thank the reviewer for this suggestion. We have conducted experiments on **Qwen2.5-7B**.
>
> **Table: Scaling Performance of EvolveR**
> | Method | Avg. Score |
> | :--- | :---: |
> | **EvolveR-instruct (0.5B)** | **0.150** |
> | **EvolveR-instruct (1.5B)** | **0.270** |
> | **EvolveR-instruct (3B)** | **0.382** |
> | **EvolveR-instruct (7B)** | **0.417** |
>
> EvolveR (7B) achieves **0.417**, significantly improving over the 3B version (0.382), proving the framework scales effectively. Crucially, we conducted an ablation study on the 7B model to compare our Self-Distillation against Teacher-Distillation (using GPT-4o-mini).
>
> **Table: Self-Distillation vs. Teacher-Distillation (7B)**
> | Distillation Method | Avg. Score |
> | :--- | :---: |
> | Teacher-Distillation (GPT-4o-mini) | 0.393 |
> | **Self-Distillation (EvolveR)** | **0.417** |
>
> As shown above, our **Self-Distillation** mechanism (0.417) outperforms **Teacher-Distillation** (0.393). This confirms that for larger models, learning from their own internal policy is more effective than relying on external teachers, highlighting the unique value of our self-evolving paradigm.

---

> ### Comment · Reviewer_bDce · 2025-11-23
>
> Thank you for the detailed and thoughtful rebuttal. I am glad that the first concern was thoroughly addressed and that additional experiments were conducted to clarify the points. I appreciate the authors’ efforts, and I will update my rating from 4 to 6. I also hope that these explanations and additional results can be included in the camera-ready version if permitted by the conference policy.

---

> > ### Author Response · Authors · 2025-11-24
> >
> > We are very grateful for your time and the decision to raise the score. We are glad that our response resolved your concerns. As suggested, we will ensure that all the additional experiments are included in the final version of the paper.

---

### Official Review · Reviewer_nkwS · 2025-10-27

**Soundness:** 4
**Presentation:** 3
**Contribution:** 3
**Rating:** 4
**Confidence:** 4

**Summary:**

EvolveR addresses the context forgetting problem in LLM agents (where each interaction is treated independently) through a self-evolution paradigm that alternates between offline self-distillation of trajectories into strategic principles and online interaction with principle retrieval. The framework employs GRPO-based policy updates and a dynamic experience base with semantic deduplication and quality scoring (Eq 2). Evaluated on seven QA benchmarks, EvolveR achieves 0.382 average score on Qwen2.5-3B, outperforming Search-R1 (0.325) and demonstrating that self-distillation surpasses teacher distillation at the 3B scale.

**Strengths:**

1.  Empirical results looks solid outperforming strong baseline like Search-R1

     a. The paper demonstrates consistent improvements across 7 diverse QA benchmarks (both in-domain and out-of-domain), with EvolveR achieving 0.382 average score on Qwen2.5-3B versus Search-R1's 0.325, showing robust generalization.

2. Ablation experiments (Table 2, 3) show that experience retrieval is crucial, however lack further analysis (see Q1). I personally like the additional of distillation experiments as well.

**Weaknesses:**

1. While the paper describes deduplication and quality control (Eq 2, line 302 ), there's insufficient analysis of long-term scalability. How does performance change as the experience base grows to thousands of principles?

2. The claim (lines 427-431) that self-distillation surpasses teacher distillation at 3B due to “cognitive alignment” is not concrete enough as it is only seen at 3B scale while smaller scales show reverse result.

3. Some incomplete analysis of limitations and failure modes:

    a. Dependency on GPT-4o for cold-start (Table 5) somewhat contradicts "self-evolving" narrative.

    b. Appendix A.4's finding that absorbing principles hurts performance deserves main paper discussion

**Questions:**

Major:
1. How does the use of different actions ( search_knowledge, search_experience, think ) change during the RL process, does search_knowledge or search_experience increase with more steps?

   a. Could you provide a longitudinal analysis to show how principle quality evolves across RL iterations?

    b. Also, please provide qualitative examples of high-scoring vs low-scoring principles to understand what makes principles effective.

Minor:

2.  Can you provide computational costs? Training time, memory requirements, and inference overhead compared to baselines would help assess practical viability.

3. In the distillation experiment at 3B scale, would scaling more data helps with the performance? I would assume the data size in the distillation is fixed for all 3 scales but for bigger models, we would expect it needed more data to perform better?

4. Although as listed in one of the limitations which stated experiments only include QA, I would still curious would non QA tasks ( as out of domain ) such as coding or multi-turn conversations benefit from EvolveR?

---

> ### Author Response · Authors · 2025-11-23
> **Response [1/2]**
>
> We thank the reviewer for the detailed and insightful comments. Below we provide point-by-point responses.
>
> # Response to Weakness 1: Long-term Scalability
>
> We thank the reviewer for raising this critical design consideration. EvolveR addresses scalability via Dynamic Quality Control (semantic deduplication and pruning of low-score principles), which prevents the experience base from growing indefinitely with redundant information.
> Regarding computational overhead, our empirical results demonstrate that retrieval remains extremely efficient even at scale. For example, with an experience base size of approximately **14,000 principles**, the latency for retrieving the top-3 principles is around **0.06 seconds**. This negligible overhead confirms that the system remains highly practical for long-term deployment.
>
>
> # Response to Weakness 2: Cognitive Alignment (Concrete Evidence from 7B Model)
>
> The reviewer rightly pointed out that the "cognitive alignment" claim needed more concrete evidence. To address this, we extended our experiments to the **Qwen2.5-7B** scale during the rebuttal.
>
> **Table: Self-Distill vs. Teacher-Distill across Scales**
>
> | Model Scale | Teacher-Distill (GPT-4o-mini) | Self-Distill (EvolveR) | **Gap** |
> | :--- | :---: | :---: | :---: |
> | **0.5B** | **0.220** | 0.150 | Teacher +0.070 |
> | **1.5B** | **0.290** | 0.270 | Teacher +0.020 |
> | **3B** | 0.370 | **0.382** | **Self +0.012** |
> | **7B** | 0.393 | **0.417** | **Self +0.024** |
>
> * Small Scale (0.5B): The base model struggles to generate coherent reasoning, so external teacher guidance is essential.
> * Large Scale (3B & 7B): As the model scales up, Self-Distill not only surpasses Teacher-Distill, but the gap widens (from +0.012 to +0.024).
>
> This concretely validates the "Cognitive Alignment" hypothesis: For capable models, principles distilled from their own policy are better aligned with their inherent reasoning style and error patterns than advice from an external model (even a stronger one).
>
>
> # Response to Weakness 3(a): Cold-Start Dependency
>
> We clarify the nature of the cold-start phase:
> *   Format vs. Strategy: GPT-4o is used *only* to generate a small set of initial trajectories to teach the agent the interaction format (e.g., how to output `<think>`, `<search_knowledge>` and `<search_experience>` tags correctly). It is not used to teach the cold-start model knowledge.
> *   One-off Process: This is a standard "warm-up" procedure in RL for LLMs to ensure the initial policy is executable. Once this brief phase is completed, the dependency on the external model is removed. All subsequent processes (including reasoning with principles, principle distillation and policy updates) are driven solely by the agent itself.
>
>
> # Response to Weakness 3(b): Experience Internalization
>
> We agree this is a valuable finding and will move the discussion from the Appendix to the main text (Section 5.3) in the revised version.
> Beyond the issue of noise from irrelevant principles, we hypothesize a mismatch in the optimization objective. Currently, unmasked experience tokens are treated identically to generated reasoning steps and optimized via GRPO's advantage-based loss. However, effective internalization might require treating high-quality principles as "ground truth" knowledge to be memorized. Therefore, a distinct loss formulation, such as likelihood maximization specifically to the `<experience>`  block, separate from the RL loss, might be necessary to properly "absorb" this wisdom without destabilizing the policy.

---

> ### Author Response · Authors · 2025-11-23
> **Response [2/2]**
>
> # Response to Question 1: Longitudinal Analysis & Qualitative Examples
> ## 1. Longitudinal Analysis of Action Usage:
>
> We analyzed the evolution of action frequencies (think, search\_knowledge, search\_experience) across RL training steps. As the reviewer asked, the usage of search actions does **not increase indefinitely**. Instead, both `<search_knowledge>` and `<search_experience>` converge to a stable range (typically 3-5 times per trajectory). This behavior is explicitly guided by our **Format Reward**, which provides a bonus for search quantity up to a cap (6 times) but does not reward excessive searching. Similarly, `<think>` converge to about 6-8 times, balancing reasoning depth with efficiency.
> In the early phase, the agent explores various action combinations. As training progresses, it learns to utilize `<search_experience>` more effectively, often prioritizing it to retrieve high-level principles that then guide subsequent `<search_knowledge>` queries, leading to a more structured and efficient problem-solving pattern.
>
> And we track the quality of principles in the Experience Base using their average `metric_score`. The average quality score shows an upward trend throughout the training process. This is driven by our dynamic curation mechanism. Newly distilled principles are initialized with a score of 0.5. As training proceeds, principles that lead to successful trajectories accumulate higher scores (up to 1.0), while ineffective ones see their scores drop. Crucially, our periodic **pruning mechanism** removes principles that fall below the threshold ($\theta_{prune}$), filtering out low-quality "noise."
>
> ## 2. Qualitative Examples of Principles:
> We provide real examples from our experience base to illustrate quality evolution.
>
> | Type | Metric Score | Content | Analysis |
> | :--- | :---: | :--- | :--- |
> | **Low Score** (Eliminated) | 0.25 | *"When using multiple principles to guide reasoning, ensure there is no redundancy or unnecessary overlapping, leading to confusion about the principle applicability and efficiency."* | **Vague:** This is a generic statement about the process, not a specific strategy for solving the task. It offers no actionable guidance. |
> | **High Score** (Retained) | 0.80 | *"When determining character relationships in a series, avoid assuming actors based solely on character recognition; confirm with verified role descriptions."* | **Specific & Actionable:** This principle identifies a specific pitfall (hallucinating actors) and provides a concrete correction strategy. |
>
>
> # Response to Question 2: Computational Costs
>
> * **Training:** The full lifecycle (SFT cold-start + RL) for the 3B model takes approximately 39.4 hours on 8 A100 GPUs.
> * **Inference:** The overhead from retrieving experience is minimal. For example, with an experience base size of approximately **14,000 principles**, the latency for retrieving the top-3 principles is around **0.06 seconds**
>
>
> # Response to Question 3: Scaling Data for Distillation
> We clarify that our experimental results represent the **optimal performance achieved before model collapse**, rather than being limited by the volume of data. Unlike offline distillation with a fixed dataset, our distillation data is generated dynamically from the agent's own rollouts during RL. The "data size" is effectively determined by the number of training steps required to reach convergence. For all model scales, we trained the agents until performance saturated. Extending the training process to force the ingestion of "more data" does not improve performance; on the contrary, we observed that it leads to model collapse (performance degradation).
>
> # Response to Question 4: Generality Beyond QA Tasks
> We acknowledge the reviewer's valid point regarding the evaluation scope. We selected these specific QA benchmarks because they require the agent to perform multi-step document search and complex reasoning to generate answers. These tasks serve as rigorous proxies for evaluating an agent's ability to plan strategically and process information. We agree that the EvolveR framework is inherently task-agnostic, and we are committed to extending it to broader domains such as coding and embodied interaction in future work to further validate its generalizability.

---

> > ### Comment · Reviewer_nkwS · 2025-11-24
> >
> > Thanks for the detailed responses, your response did indeed fill up missing gaps from your work. The 7B experiments for cognitive alignment (W2) are excellent and significantly strengthen your claim.
> >
> > However, I have concerns about incomplete responses to my original questions and conceptual issues with some explanations provided.
> >
> > >  we track the quality of principles in the Experience Base using their average metric_score. The average quality score shows an upward trend throughout the training process.
> >
> > This metric is confounded by selection bias and doesn't answer my question about whether the agent improves during training process.  Specifically:
> >
> > 1. Since you're pruning principles with scores < 0.3 (lines 233-238, 679-680), this procedure tends to inflate the average by construction, even without genuine learning. This observation is simply a mechanism you designed working, not evidence of learning.
> >
> > 2. What I really asking is that : Does the agent get better at distilling principles over time? To demonstrate this, you need to show that principles distilled at iteration 500 are higher quality than those distilled at iteration 100, controlling for number of uses.
> >
> > As for the major Q1: Longitudinal Analysis response, I think you have some rough numbers, but I would like to include it in appendix showing the reader what the model actually doing during inference.
> >
> > I'm willing increase my score if you:
> >
> > 1. Address the "average quality" metric issue with proper analysis
> >
> > 2. Add the longitudinal action evolution analysis to the main paper (Which I suspect you have most of the data required for the analysis)

---

> > > ### Author Response · Authors · 2025-11-26
> > >
> > > We are grateful for the reviewer's positive feedback on our 7B experiments and the clear suggestions. We have addressed the two remaining concerns as follows:
> > >
> > > ## 1. Addressing "Selection Bias": Evidence of Principle Quality Improvement
> > >
> > > We agree that the average score of the entire pool is confounded by pruning. To isolate the agent's **intrinsic principle distillation capability**, we analyzed the quality of principles grouped by their distilled time.
> > >
> > > **Table: Quality of Principles by steps**
> > >
> > > | **Creation Phase** | **Avg. Metric Score** | **Avg. Usage Count**
> > > | :--- | :---: | :---: |
> > > | Interval 1 (step 1～15) | 0.462 | 8.22 |
> > > | Interval 2 (step 15～30)| 0.464 | 24.10 |
> > > | Interval 3 (step 30～45)| 0.487 | 17.82 |
> > > | Interval 4 (step 45～60)| **0.500** | **12.36** |
> > >
> > > Interpretation：
> > > *  **Interval 1:** These principles have undergone the longest period of pruning. They represent the high quality principles of the early phase.
> > > *  **Interval 4:** These are generated at the later stage of training. Despite being newer and undergoing **less filtering** than Interval 1, they achieve a **higher average score (0.500 > 0.462)**.
> > >
> > > If the agent were not learning, the newer principles should score lower than the filtered older ones. The fact that they score higher—with substantial usage counts (>12) confirming stability—proves that the agent is generating fundamentally higher quality strategies in later stages.
> > >
> > >
> > > ## 2. Longitudinal Action Analysis
> > > We analyzed the evolution of action frequencies across training steps. We will include the full plots in the revised version. Here, we summarize the trend in a table divided into four training phases.
> > >
> > > **Table: Average Action Counts across Training Phases**
> > >
> > > | **Training Phase** | **Search Experience** | **Search Knowledge** | **Think**|
> > > | :--- | :---: | :---: | :---: |
> > > | Interval 1 (step 1～15) | 2.02 | 2.10 | 4.10 |
> > > | Interval 2 (step 15～30)| 2.15 | 3.20 | 5.50 |
> > > | Interval 3 (step 30～45)| 2.05 | 3.95 | 6.15 |
> > > | Interval 4 (step 45～60)| 2.12 | 4.05 | 6.20 |
> > >
> > >
> > > Thanks to the cold-start stage, the agent exhibits reasonable tool usage capabilities even in the early phase. As RL training progresses, we observe a clear optimization trend: the frequencies of `<think>` and `<search_knowledge>` gradually increase to support more complex reasoning, while `<search_experience>` remains relatively stable.
> > >
> > > Crucially, all action counts converge to a stable range in the final phase (approx. 2 experience searches, 4 knowledge searches, and 6 thinkings). This convergence confirms that the agent guided by the format reward and its evolving policy, and establishes a consistent, efficient problem-solving pattern.

---

> > > > ### Comment · Reviewer_nkwS · 2025-11-27
> > > >
> > > > Thanks for the detailed response, I have adjusted my score accordingly.

---

> > > > > ### Author Response · Authors · 2025-11-27
> > > > >
> > > > > We are very grateful for your time and the decision to raise the score. We are glad that our responses and additional experiments resolved your concerns. We will ensure that all the new analyses are included in the revised version of the paper.

---

### Official Review · Reviewer_AKYC · 2025-10-31

**Soundness:** 2
**Presentation:** 3
**Contribution:** 3
**Rating:** 4
**Confidence:** 3

**Summary:**

The paper proposes EvolveR, a closed-loop lifecycle for agent self-evolution with two alternating stages: (i) Offline Self-Distillation that converts raw trajectories into reusable principles with deduplication and dynamic scoring, and (ii) Online Interaction where the agent retrieves principles to guide reasoning; the loop is closed with policy evolution via RL (GRPO). Experiments on seven QA datasets (NQ, HotpotQA, TriviaQA, PopQA, 2Wiki, Musique, Bamboogle) using Qwen2.5 backbones show gains over prompting, SFT, and RL baselines; ablations analyze self-distill vs teacher-distill and the role of principle retrieval.

**Strengths:**

1.	Clear lifecycle design that operationalizes “experience as principles” rather than raw cases; includes deduplication and dynamic scoring for maintenance.
2.	Integrated RL (GRPO) ties the use of retrieved principles to policy updates via a composite reward (outcome + format), providing a concrete optimization path.
3.	Broad evaluation across seven QA benchmarks with competitive improvements; main table shows best average at 3B.
4.	Insightful ablations.

**Weaknesses:**

1.	All experiments are QA-centric; it is unclear whether principle distillation generalizes to planning, tool orchestration, or embodied tasks. The paper frames a general “agent evolution” paradigm but evaluates only text QA.
2.	The format reward is hand-crafted with specific thresholds (e.g., counts of <think>/<search>), which could bias behaviors; robustness to alternative reward shapes or automatic shaping is not shown.
3.	The score s(p)= (csucc+1)/(cuse+2) is simple; its sensitivity to retrieval noise, task difficulty, or distribution shift is not analyzed. Limited justification for thresholds (θ_sim, θ_prune) and no calibration study.
4.	Table 1 averages improve (0.382), but per-dataset effect sizes vs best baselines are sometimes modest; variance/error bars, seed sensitivity, and cost/throughput are not systematically reported.
5.	Limited analysis of failure modes Although examples/prompts are in Appendix, the main paper provides few qualitative failures where principles mislead or conflict; Appendix notes a negative result for “exp-absorb,” but broader diagnostics remain thin.

**Questions:**

1.	Can you show results on non-QA agent tasks (multi-step tool use, program synthesis, web navigation, or planning) to validate the paradigm beyond QA?
2.	How sensitive are results to θ_sim, θ_prune, and the format reward hyperparameters? Any ablation on these knobs?

---

> ### Author Response · Authors · 2025-11-23
> **Response [1/2]**
>
> We sincerely thank the reviewer for the detailed and constructive feedback. We appreciate the recognition of our "clear lifecycle design" and "insightful ablations." We address your concerns regarding generalization, robustness, and experimental details below.
>
> # Response to Weakness 1 & Question 1: Generality Beyond QA
>
> We acknowledge the reviewer's valid point regarding the evaluation scope. We selected these specific QA benchmarks because they act as rigorous proxies for **general reasoning and planning capabilities**.
> 1. **Complexity:** Tasks in HotpotQA and 2WikiMultiHop require the agent to perform **multi-step planning**, decompose complex goals into sub-queries, and execute precise tool calls. These are fundamental capabilities for any generalist agent, extending beyond simple fact retrieval.
> 2. **Future Work:** We agree that the EvolveR framework is inherently task-agnostic, and we are committed to extending it to broader domains such as coding and embodied interaction in future work to further validate its generalizability.
>
>
>
> # Response to Weakness 2: Format Reward Robustness
>
> We clarify the design philosophy of the format reward:
> *  **Syntax Constraint, Not Semantic Guide:** The format reward is designed solely to ensure the model adheres to the correct interaction protocol (e.g., generating valid `<think>`, `<search_knowledge>` and `<search_experience>` tags).
> *  **Minimal Bias:** We deliberately kept this reward lightweight and task-agnostic. The actual "behavior shaping"—learning **what to search and how to reason**—is driven by the **Outcome Reward** (correctness) via GRPO. This separation ensures that the agent optimizes for task success rather than hacking the reward function, a standard practice in recent RL-for-LLM works (e.g., Search-R1[1]).
>
> [1] Jin B, Zeng H, Yue Z, et al. Search-r1: Training llms to reason and leverage search engines with reinforcement learning[J]. arXiv preprint arXiv:2503.09516, 2025.
>
>
> # Response to Weakness 3 & Question 2: Sensitivity of Scoring and Thresholds
>
> We thank the reviewer for this suggestion. To demonstrate the robustness of our system, we conducted a sensitivity analysis on the pruning threshold $\theta_{prune}$ using the Qwen2.5-1.5B model.
>
> **Table: Sensitivity Analysis of $\theta_{prune}$ (Qwen2.5-1.5B)**
>
> | **$\theta_{prune}$** | 0.1 (Loose) | 0.3 (Default) | 0.7 (Strict) | 0.9 (Very Strict) |
> | :--- | :---: | :---: | :---: | :---: |
> | **Avg. Score** | 0.269 | 0.270 | 0.265 | 0.276 |
>
> As shown in the table, the performance remains remarkably stable (fluctuating between 0.265 and 0.276) across a wide range of thresholds. $\theta_{prune}=0.3$ (Default) provides a good balance. Surprisingly, a very strict threshold ($\theta_{prune}=0.9$) also performs well (0.276), suggesting that retaining a small set of "elite" principles is highly effective. A very loose threshold ($\theta_{prune}=0.1$) leads to a larger experience base with slightly more noise, but performance (0.269) does not degrade significantly. This confirms that EvolveR is robust to hyperparameter variations and does not rely on precise tuning of $\theta_{prune}$ to function.

---

> ### Author Response · Authors · 2025-11-23
> **Response [2/2]**
>
> # Response to Weakness 4: Effect Sizes and Computational Cost
>
> 1. **Effect Sizes:** While gains on simple retrieval tasks are modest due to saturation, EvolveR achieves a substantial 17.5% relative improvement overall compared to Search-R1-instruct (0.382 vs. 0.325). Crucially, our advantage is magnified on complex reasoning benchmarks like Musique (+33%) and Bamboogle (+24%), confirming that our framework excels at learning strategic reasoning where baselines struggle.
> 2. **Seed Sensitivity:**: Due to the high computational cost of RL training on LLMs (approx. 39.4 hours 8 A100 GPUs for per 3B experiment), conducting multi-seed variance analysis for every baseline is computationally prohibitive.
> 3. **Computational Cost:** We have added a detailed cost analysis and will add them in the Appendix:
>     * **Training:** The full lifecycle (SFT + RL) for the 3B model takes approx. 39.4 hours on 8 A100 GPUs.
>     * **Experience Retrieval Latency:** The retrieval overhead is negligible. With an experience base size of approximately **14,000 principles**, the latency for retrieving the top-3 principles is around **0.06 seconds** which is imperceptible compared to LLM generation time.
>
>
> # Response to Weakness 5: Failure Modes
>
> We agree that analyzing failures is crucial. We will moved the discussion of **"Experience Internalization"** (gradient unmasking) from the Appendix to the main text (**Section 5.3**) in the revised version.
>
> We explicitly discuss why directly absorbing experience (unmasking gradients) leads to performance degradation. We attribute this to "training noise" from imperfect retrieval. Beyond the issue of noise from irrelevant principles, we hypothesize a mismatch in the optimization objective. Currently, unmasked experience tokens are treated identically to generated reasoning steps and optimized via GRPO's advantage-based loss. However, effective internalization might require treating high-quality principles as "ground truth" knowledge to be memorized. Therefore, a distinct loss formulation, such as likelihood maximization specifically to the <experience><\experience> block, separate from the RL loss, might be necessary to properly "absorb" this wisdom without destabilizing the policy.

---

> ### Author Response · Authors · 2025-11-27
>
> Thank you once again for your valuable review. In our rebuttal, we have provided the detailed hyperparameter sensitivity analysis and computational cost report to address your concerns about robustness and efficiency.
>
> With the end of the discussion phase approaching, we would appreciate it if you could let us know if these updates resolve your concerns. We are happy to provide any final clarifications if needed.

---

### Meta-Review · Area_Chair_64jf · 2026-01-07

**Summary:**

The paper proposes a self-improving agent loop by distilling trajectories into principles, retrieving them online, then updating with GRPO. The writing is clear and the QA experiments look reasonably solid, and the rebuttal helps with the added 7B results and the RL vs SFT ablation. However, almost everything is still QA-only, so it’s hard to buy the “general agent” framing without at least one non-QA setting (tool use, coding, web, etc.). The idea is clear and effective, but the validation feels too narrow right now. I would recommend rejection.

**Reviewer Concerns:**

- Rebuttal addressed: (1) provided additional evidence and clarifications on principle quality improvement and added longitudinal action-usage analysis, (2) strengthened the “cognitive alignment” claim with larger-scale results and clearer comparisons to teacher distillation, (3) added practical efficiency/cost details and clarified positioning vs. concurrent methods.

- Still outstanding: (1) generalization beyond QA remains unvalidated, (2) robustness analyses are still partial (limited sweeps...), (3) scalability/lifelong behavior is argued via pruning and latency but lacks longer-horizon stress tests or qualitative failure diagnostics in the main paper.

**Reviewer Scores:**

- AKYC: slightly positive, but still constrained by the QA-only evaluation and limited robustness reporting.
- nkwS: slightly positive, since the rebuttal directly addressed the confound and added the requested longitudinal analysis.
- bDce: positive, after clearer positioning vs. related systems plus added scaling/cost evidence.
- J5YF: maybe the same, added RL-vs-SFT ablation and larger-model results, but broader validation remains limited.

---

### Decision · Program_Chairs · 2026-01-26

Reject